# Enhancing Consistency of Microphysical Properties of Precipitation across the Melting Layer in the Dual-Frequency Precipitation Radar Data

Kamil Mroz[1], Alessandro Battaglia[2], and Ann M. Fridlind[3]

[1]National Centre for Earth Observation, University of Leicester, Leicester, UK
[2]Department of Environment, Land and Infrastructure Engineering (DIATI), Politecnico of Turin, Turin, Italy
[3]NASA Goddard Institute for Space Studies, New York, NY, USA

**Correspondence:** Kamil Mroz (kamil.mroz@le.ac.uk)

**Abstract.** Stratiform rain and the overlying ice play crucial roles in the Earth's climate system. From a microphysics standpoint, water mass flux primarily depends on two variables: particles concentration and their mass. The Dual-frequency Precipitation Radar (DPR) on the Global Precipitation Measurement mission core satellite is a space-borne instrument capable of estimating these two quantities through dual-wavelength measurements. In this study, we evaluate bulk statistics on the ice particle prop-
erties derived from dual-wavelength radar data in relation to the properties of rain underneath. Specifically, we focus on DPR observations over stratiform precipitation, characterized by columns exhibiting a prominent bright band, where the melting layer can be easily detected.

Our analysis reveals a large increase in the retrieved mass flux as we transition from the ice to the rain phase in the official DPR product. This observation is in disagreement with our expectation that mass flux should remain relatively stable across the
bright band in cold rain conditions. To address these discrepancies, we propose an alternative retrieval algorithm that ensures a gradual transition of $D_m$ (mean mass-weighted particle melted-equivalent diameter) and precipitation rate across the melting zone. This approach also helps in estimating bulk ice density above the melting level. These findings demonstrate that DPR observations can not only quantify ice particle content and their size above stratiform rain regions but also estimate bulk density, provided uniform conditions that minimize uncertainties related to partial beam filling.

*Copyright statement.* TEXT

## 1   Introduction

As part of the Global Precipitation Measurement (GPM; Hou et al., 2014) mission, in February 2014, the US and Japan National Agencies jointly launched the space borne Dual-frequency Precipitation Radar (DPR). The DPR measures at Ku (13.6 GHz) and Ka (35.5 GHz) bands, improving on the single-frequency Ku-band (13.8-GHz) radar that was launched in 1997 with the
Tropical Rainfall Measuring Mission (TRMM; Kummerow et al., 1998). Not only does GPM have an extended latitude range

of 65°S–65°N, compared to TRMM's 36°S–36°N, but it has also improved the system sensitivity. In fact, the Ku Precipitation Radar is now able to detect a minimum radar reflectivity of 15.5 dBZ (Liao and Meneghini, 2022), whereas during the TRMM era, it was 18 dBZ (Heymsfield et al., 2000). Thanks to extended latitudinal coverage, the DPR shown that the world's largest storms occur primarily over mid- and high-latitude oceans in both hemispheres (Liu and Zipser, 2015), and that the areal coverage of storms overshooting the tropopause over central North America and Argentina rivals that over the tropics (Liu and Liu, 2016).

Here we investigate use of DPR data for the retrieval of the size and the precipitation rate of ice crystals overlying cold rain. Widespread precipitation associated with convection is a primary source of stratiform rain, especially in the tropics. Observational and modelling studies have focused on this topic for decades (e.g., Webster and Stephens, 1980; Chen and Cotton, 1988). However, the climatological significance of fundamentally differing diabatic heating profiles corresponding to regions of convective and stratiform rain was fully realized only about 20 years ago (Houze, 1997). Since then, TRMM measurements have enabled us to assess the frequency and extent of stratiform rain in tropical and subtropical regions. This classification has unveiled the existence of more extensive stratiform rain regions over the ocean, even though the embedded convection is notably more intense over land (Houze et al., 2015; Schumacher and Funk, 2023). Today, identification of precipitation type as convective, stratiform or neither is a leading element of DPR retrievals (Awaka et al., 2021).

The microphysical properties of ice particles overlying stratiform rain have a strong impact on local rain rate and the size and duration of stratiform regions (e.g., Jensen et al., 2018). However, the physical processes governing the size and evolution of ice particles are still poorly understood (Ackerman et al., 2015; Barnes and Houze Jr., 2016; de Laat et al., 2017; Fridlind et al., 2017; Ladino et al., 2017; Lawson et al., 2017), highlighting the need for improved observational data sources to constrain model physics. While ice physics within convective columns is challenging to observe, stratiform rain decks are widespread and long-lived features that can be robustly observed and evaluated in simulations (e.g., via aircraft in situ measurements, Fridlind et al., 2017). The question is, to what extent can satellite remote-sensing platforms be used to retrieve physically robust ice properties within these features? Single-wavelength radar reflectivity cannot reliably constrain ice size and concentration due to non-unique signatures from each variable (Drigeard et al., 2015). The CloudSat radar and Cloud-Aerosol Lidar and Infrared Pathfinder Satellite Observations (CALIPSO) introduced two-frequency (two-instrument) retrievals above the lidar extinction level (Delanoë and Hogan, 2010; Deng et al., 2010), providing an effective technique for characterizing cirrus clouds and the tops of thicker clouds. Thanks to its weaker attenuation the GPM radar offers an opportunity to characterize the microphysical properties of precipitable size ice particles based on well-colocated dual-frequency measurements. The possibility to characterize ice properties globally over stratiform rain regions from a space-borne sensor provides a unique opportunity for advancing understanding of fundamental weather processes and offers a valuable constraint for improving climate model physics.

In the following sections, we first provide an example of DPR data obtained from a deep stratiform rain volume, and compile five years of global statistics on the DPR observables and the retrieved microphysical parameters in proximity to the melting layer (Sect. 3.2). Based on these statistics, some physically inconsistent properties of the precipitation columns are highlighted. Then, we demonstrate an alternative retrieval scheme for ice crystal properties that differs from the official DPR algorithm in the

ice crystal density assumptions (Sect. 4). It should be noted that this algorithm is only applicable to homogenous precipitation columns and where both frequency measurements exceed the radar sensitivity threshold. The new retrieval is validated with the polarimetric radar product in Sect. 6. Results and conclusions are summarized in Sect. 7.

## 2    DPR measurements

To showcase the measurement capabilities of the DPR system, we present an example of its observations and follow it with statistics on precipitation properties in the vicinity of the melting layer.

### 2.1    Example scene

Figure 1 provides an example of level 2 Ku- and Ka-band measurements from a deep stratiform outflow region over Missouri in June 2015. The stratiform rain is associated with widespread Ku-band reflectivity values peaking around $\sim 40$ dBZ below

the melting level. For the Ka-band channel, much weaker echoes are observed as a result of the cumulative attenuation effects, mainly caused by raindrops and melting snowflakes, and non-Rayleigh scattering signatures. In this example, the stratiform rain is likely associated with the outflow from the squall line system that can be seen at the southern-east end of the scene.

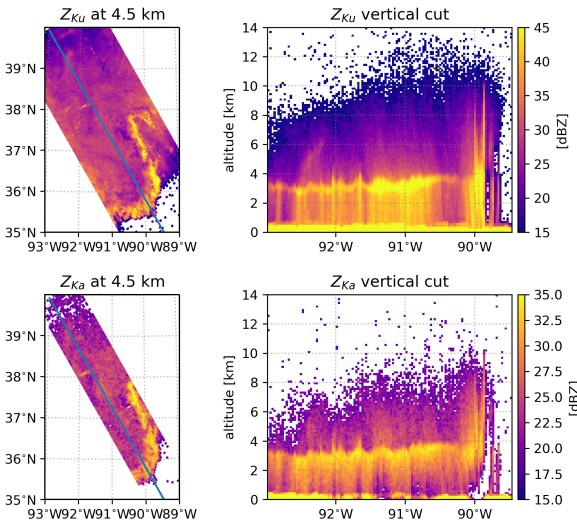

**Figure 1.** DPR Ku- and Ka-band radar reflectivity measurements of a storm over Missouri on 26th June 2015 (orbit 18010). The panels on the left depict the horizontal cross-section at the constant altitude of 4.5 km ASL, whereas the panels on the right show the vertical cross-cut along the satellite track (ray 29th).

The vertical cross-section at the Ku-band along the satellite track shows a pronounced bright band at 3-4 km above sea level (ASL), which is characteristic of stratiform rain observed by low frequency radars. Corresponding Ka-band vertical cross-

sections do not show a bright band but exhibit a sharp increase in radar reflectivity due to the water phase transition and the

resulting change in dielectric constant. It is evident that the Ku-band system is more sensitive than the Ka-band radar, with a sensitivity of 15.5 dBZ compared to 19.2 dBZ (Liao and Meneghini, 2022). Therefore, it is capable of detecting ice particles above 12 km in elevation for the presented weather system. In addition to distinct sensitivity levels, the measured reflectivity at the two bands above the melting zone exhibit a large difference. The measured reflectivity is a combination of the effective

reflectivity, $Z_e$, of the distributed hydrometeors at range $r$ and the 2-way path integrated attenuation (PIA):

$$Z_m(r) = Z_e(r) - 2PIA(r) = Z_e(r) - 2\int_0^r k(s)\,\mathrm{d}s, \tag{1}$$

where $k$ is the one-way specific attenuation along the path. Therefore, the difference in measured reflectivity, so-called the Dual-Frequency wavelength Ratio (DFR):

$$\mathrm{DFR}(r) = Z_m^{Ku}(r)[\mathrm{dBZ}] - Z_m^{Ka}(r)[\mathrm{dBZ}] \tag{2}$$

$$= Z_e^{Ku}(r) - Z_e^{Ka}(r)+ \qquad\qquad \text{(non-Rayleigh)}$$

$$2\left[\mathrm{PIA}^{Ka}(r) - \mathrm{PIA}^{Ku}(r)\right] \qquad\qquad \text{(differential attenuation)}$$

is a result of differences in $Z_e$ (non-Rayleigh effects) or/and differences in specific attenuation at different channels (differential attenuation). The non-Rayleigh effects are most evident just above the melting layer where particles tend to aggregate and form large ice particles, while attenuation effects are not as strong (see e.g. Kuo et al., 2016). For the scene shown in Fig. 1, the DFR

reaches 7 dB in the ice phase. Below the melting zone, the DFR typically increases towards the surface due to the dominant effects of differential attenuation caused by rain.

## 2.2 Measurement and retrieval statistics

The statistics presented in this section are based on an analysis of DPR data collected between 2015 and 2020. We selected only stratiform rain columns for analysis, based on the following criteria. First, precipitation was detected by both GPM radars

(flagPrecip=11 in the DPR Level 2 data files, for more details see the documentation file prepared by JAXA/EORC Team, 2017). Second, a bright band was detected at the Ku-band (flagBB=1) with a high quality of the detection (qualityBB=1). Third, the stratiform rain type was identified (the first bit of typePrecip equal to 1) with high certainty (qualityTypePrecip=1). And fourth, to avoid any contamination with melting particles introduced by the slanted geometry of the scan, only five central scans along the satellite track that correspond to almost nadir measurements are selected. Note that the DPR beam has an

approximate width of 5 km. As a result, the discrepancy in the altitude of the bottom boundary of the radar bin can reach approximately 1500 m (for Ku-band) or 700 m (for Ka-band) at the edge of the swath where the local zenith angle is equal to $17°$ or $8°$, respectively. Moreover, before the swatch structure change in May 2018 (Liao and Meneghini, 2022), Ku- and Ka-band radar data were not matched in the outer swath and, consequently, dual-frequency retrieval was not available in that region.

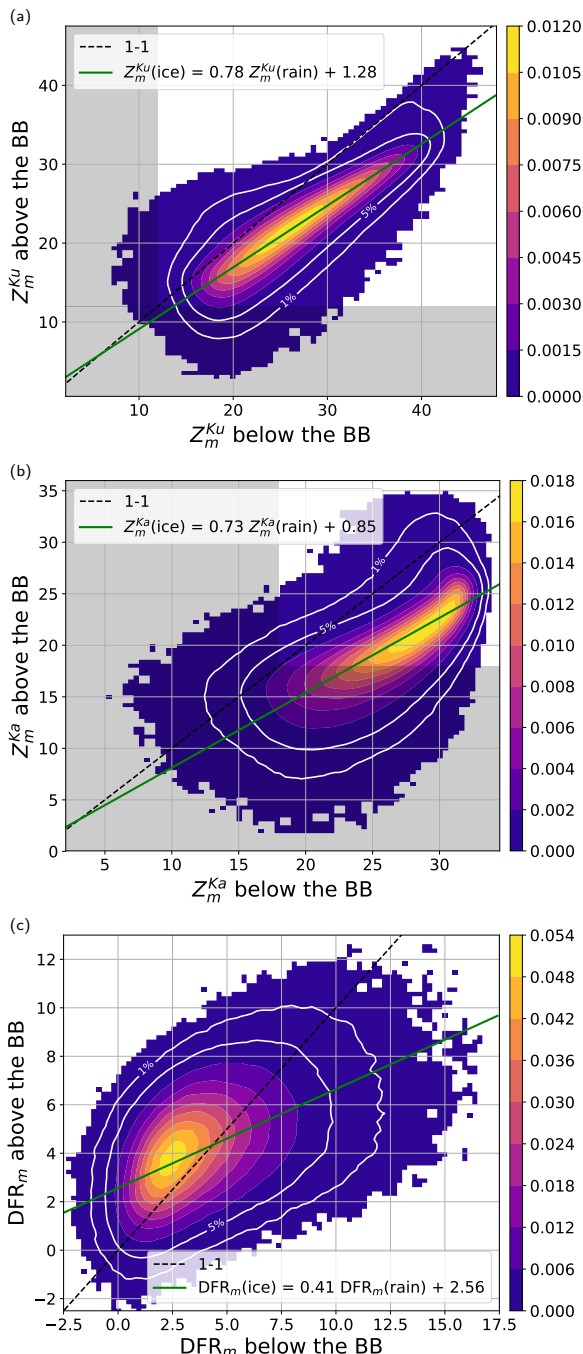

**Figure 2.** The joint probability distribution functions (PDFs) of the DPR observables measured above and below the melting zone. The green line represents a linear fit passing through the most commonly observed values for a given argument $x$. The grey shaded area represents measurements below the signal-to-noise ratio of 1. The white contour lines indicate a boundary of the regions that contain 1 and 5% of the least commonly observed values.

The aim of this study is to establish a statistical relationship between DPR observables below and above the melting zone within stratiform rain, and we focus here exclusively on observables in these two regimes. The altitude and extent of the melting zone are determined using the DPR V06 bright band detection product (Le and Chandrasekar, 2013). In order to reduce the impact of signal fluctuations caused by instrument noise, we fit the measurements from the DPR system with a first-order polynomial over a 0.5 km layer above and below the bright band, and then evaluate these polynomials at the melting layer boundaries.

Figure 2 illustrates a joint histogram of DPR observables below and above the melting zone, corresponding to the entire dataset of $\sim 9$ million columns. In general, liquid phase hydrometeors are more reflective than ice particles at both DPR channels. The difference in Ku-band reflectivities between the ice and rain phases is typically less than 15 dB. This wide range of reflectivity values for solid particles producing the same radar signal as rain highlights the diverse range of ice properties. While interpreting this signal is not straightforward, some factors that contribute to the magnitude of the reflectivity change through the melting zone can be quantified. Drummond et al. (1996) have shown (their eq. 5) that under constant mass flux assumption and Rayleigh scattering regime the reflectivity in ice is directly related to the reflectivity in rain via the following formula:

$$Z^i[\text{dBZ}] = Z^r[\text{dBZ}] + 10\log_{10}(V_D^r/V_D^i) - 6.4, \tag{3}$$

where superscripts $r$ and $i$ denote "rain" and "ice", respectively while $V_D$ is the backscattering weighted mean sedimentation velocity of particles in the radar volume. The factor of -6.4 dB corresponds to changes in the dielectric factor when water transition from solid to liquid phase (Fabry and Zawadzki, 1995). The second factor accounts for the change in particle concentration required to sustain the mass flux. Depending on the ice aerodynamics, this can compensate the scattering efficiency factor by up to 6 dB, with dense and thus fast falling particles characterized by a small reduction. The contribution from other phenomena like the orientation, shape and density of ice particles is very difficult to quantify, and it requires certain assumptions on particles' morphology. An additional complexity originates from microphysical processes that occur within the melting zone. The melting process cools down the air around particles, which induces condensation of the saturated air and an increase of the mass of particles. Furthermore, the presence of a wet coating on falling snowflakes makes them prone to aggregation within the BB (Mitra et al., 1990). On the other hand, for strongly asymmetric particles, fragmentation can also occur. The regression line that passes through the most frequently observed pairs of Ku measurements suggests that the reflectivity difference between different water phases increases with rainfall intensity. This phenomenon can be attributed to greater ice density at higher precipitation rates, which reduces the second term in formula (3). Alternatively, non-Rayleigh scattering effects in the ice portion can lead to $Z^i$ values that are lower than predicted by Drummond et al. (1996). Additionally, continuous particle growth within the melting zone can favour reflectivities higher than anticipated below the melting zone

Due to non-Rayleigh scattering, the Ka-band histogram is more compact than for the lower frequency channel. The range of observed Ka-reflectivity in ice and liquid phase are smaller by 10 dB. Moreover, the number of pairs above the 1-1 line is also higher than at 13 GHz. The characteristic hook present for strong echoes may result from non-Rayleigh scattering in rain or strong attenuation induced by ice and melting particles, which greatly reduces measured echoes below the BB. Because of

that, a wide range of radar reflectivity values in ice are observed for the same reflectivity in rain, e.g. for $Z_m^{Ka} = 30$ dBZ in rain the reflectivity in ice spans from 15 to 33 dBZ. Note that the lower end of the measured reflectivity ($Z_m^{Ka} < 18$ dBZ) must be interpreted with caution because the signal-to-noise ratio is small there.

DFRs below the BB exhibit a range between -2 and 15 dB, with most of the variability (from -1 to 10 dB) attributed to Mie scattering at the Ka-band (Seto et al., 2013), which is associated with drops larger than approximately 0.8 mm. The extreme observed DFR values arise primarily from the effects of differential attenuation, with a minor contribution stemming from random instrument noise at the Ku and Ka bands. It is worth noting that, on average, the DFR of ice tends to increase with the DFR of rain. This aligns with the intuition that larger snowflakes aloft generate bigger drops. Having said that, the same degree of variability in the DFR is observed within ice across all DFR values in rain. This suggests that droplet size is not solely dependent on the size of ice particles. Other factors, such as ice density, should be considered for accurate modelling of the measured reflectivity above the 0° isotherm.

## 3  DPR retrieval: current status

For liquid phase particles, the DPR retrieval adopts the gamma distribution model, i.e.:

$$N(D) = N_w \frac{6(\mu+4)^{\mu+4}}{4^4 \Gamma(\mu+4)} \left( \frac{D}{D_m} \right)^\mu \exp \left[ \frac{-(\mu+4)D}{D_m} \right] \tag{4}$$

and $\mu$ is assumed to be constant and equal to 3. Mixed and solid phase particle are described in terms of melted hydrometeors with the same gamma function (Iguchi et al., 2018) model. By imposing the analytic form, the description of the complex shape of the particle size distribution is reduced to two parameters only: the mean mass-weighted melted-equivalent diameter, $D_m$ [mm], and the generalized intercept parameter, $N_w$ [mm$^{-1}$m$^{-3}$]. The parameter $N_w$ functions as the scaling factor for particle concentration, while $D_m$ serves as the descriptor for the overall size of particles present within the radar volume.

The definition of ice particle size remains a subject of ongoing debate within the remote sensing community (von Lerber et al., 2017). Depending on specific application requirements, various definitions are employed to characterize the size of individual ice particles. Even within the realm of in-situ measurements, there is no universal consensus on a single definition. To illustrate this diversity, consider the example of a black and white image of a snowflake. In such cases, the size can be described using multiple criteria, including:

 – The Distance Between Furthest Pixels: This measures the greatest separation between any two points on the particle's perimeter.

 – Diameter of a Circumscribed Circle: This refers to the diameter of the smallest circle that fully encloses the snowflake.

 – Maximum Extent Along X or Y Axis: It represents the greatest length along either the horizontal (x) or vertical (y) axis of the particle.

 – Area Equivalent Circle Diameter: This method involves determining the diameter of a circle with the same area as the particle.

– Diameters of a Circumscribed Ellipse: The diameter of the smallest ellipse that fully encloses the snowflake.

     – etc...

The choice of which definition to use often depends on the specific goals and constraints of a given research or measurement task, highlighting the multifaceted nature of ice particle size characterization in remote sensing and in-situ observations.

In our current study, we choose to define the size of snow particles based on the diameter of a spherical raindrop with
equivalent mass. Consequently, $D_m$ is also expressed in terms of the melted equivalent size. Throughout the manuscript, we will refer to $D_m$ as the "characteristic size" or simply "size" for brevity. This choice aligns with the one-to-one correspondence implied by the melting process, which is a central hypothesis in our investigation. Given that the dataset of snowflake scattering properties used in this study (available at https://zenodo.org/record/7510186) also includes information on particle extent (i.e., the distance between the furthest points), it allows for a straightforward conversion from melted equivalent size to physical
size. The DPR algorithm utilizes a different snow morphology assumption. Snow is modeled as spherical particles consisting of an ice and air mixture with a density of $0.1\,\mathrm{g\,cm^{-3}}$, irrespective of the snowflake size.

The DPR retrieval algorithm utilizes measured radar reflectivity, total path integrated attenuation estimates corrected for non-precipitating particles, the relationship between precipitation rate and mass-weighted mean diameter ($PR - D_m$), and phase information based on the melting layer detection. It generates profiles of precipitation rate and drop size distribution
parameters ($D_m$, $N_w$). Additionally, profiles of effective reflectivity and specific attenuation coefficients are provided. The algorithm employs the $PR - D_m$ relationship with an adjustment parameter, $\epsilon$, aiming to reconcile discrepancies between the surface reference technique PIA and the one simulated from hydrometeor profiles. Version 06 had a single $\epsilon$ value along the profile, while Version 07 introduces varying $\epsilon$ in the column.

The $PR\text{-}D_m$ relation, replaces the traditionally used relation involving specific attenuation ($k$) and effective radar reflectivity
factor ($Z_e$). While using the $k\text{-}Z_e$ relation with the Hitschfeld-Bordan attenuation correction method (Hitschfeld and Bordan, 1954) enables the derivation of a $Z_e$ profile from the $Z_m$ profile without the need for scattering tables, this relation is not applicable at the Ka-band due to the weaker correlation between involved parameters. This limitation arises from rain extinction being strongly affected by absorption rather than being dominated by scattering. Consequently, the Hitschfeld-Bordan method leads to inconsistencies in attenuation correction at two frequencies.

The algorithm follows a logical sequence: assuming a gamma DSD with a fixed shape parameter, a relationship between $PR$ and $D_m$ imposes a unique solution for a given effective reflectivity. Consequently, the corresponding values for $N_w$ is found and by using the scattering tables the specific attenuation coefficient $k$ is obtained. The process begins at the top, where the measured reflectivity is assumed to be unaffected by attenuation and is iteratively corrected using the estimated $k$. This procedure is applied throughout the column, resulting in the attenuation profile. The process is iterated with different values of
$\epsilon$ to minimize the difference between the simulated PIA at the SRT-estimate.

For more details about the changes introduced in version 6 of the GPM-DPR algorithm, refer to the Algorithm Theoretical Basis Document (Iguchi et al., 2018) or to the algorithm description provided by Seto et al. (2021). Additionally, the study conducted by Chase et al. (2020) provides a thorough evaluation of the $PR\text{-}D_m$ relation in both rain and snow using disdrom-

eter measurements. They conclude that the $PR$-$D_m$ retrieval may not be optimal in snow due to the variability of snowflake mass, suggesting the exploration of alternative techniques.

### 3.1 Example scene

Figure 3 illustrates the DPR retrieval of the PSD parameters for the storm depicted in Figure 1. In this study we use DPR L2 version 6 (V06A) algorithm (2A.GPM.DPR files). In the solid phase region, the retrieved parameters generally show a decrease in $D_m$ and $N_w$ as the height above the bright band increases. Notably, sharp horizontal discontinuities in the values of the retrieved intercept parameter indicate that the retrieval algorithm is run for each ray separately. Additionally, the retrieval results reveal that there is a higher concentration of particles when their characteristic size is small, and vice versa. This was already observed by Chase et al. (2021) in their Fig. 15. The relationship between $D_m$ and $N_w$ varies from column to column, resulting in horizontal inconsistencies in the microphysical properties. This inconsistency is particularly evident at the detectable cloud top, where a range of sizes and particle concentrations is observed for the same radar reflectivity value.

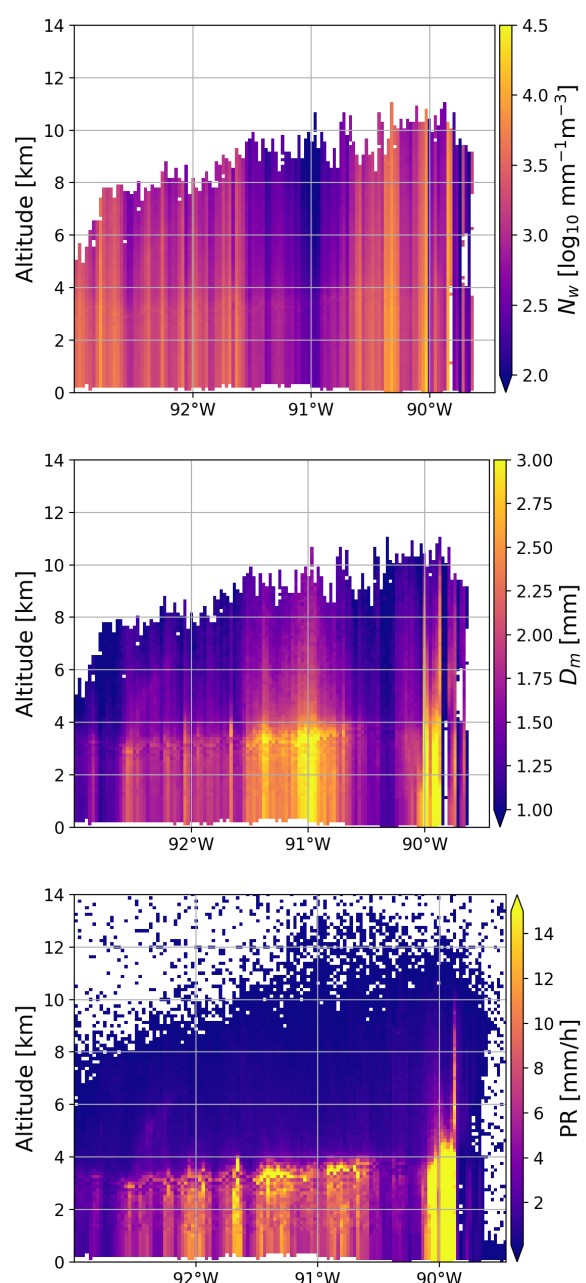

**Figure 3.** The generalized intercept parameter ($N_w$), mean mass weighted melted diameter ($D_m$) and precipitation rate as retrieved in the DPR version 6 product for the storm depicted in Fig. 1.

By assuming the falling velocity of particles is a function of diameter and density, as Seto et al. (2013), the retrieved PSD parameters can be converted into snowfall and rainfall rate. The rainfall product (Figure 3 bottom-left) clearly indicates different

characteristic features of the squall line system (Biggerstaff and Houze, 1991): the leading convective part at the southern edge of the system characterized by heavy rain; the trailing stratiform region with moderate rain intensities ($< 10$ mm h$^{-1}$) in the northern part of the system and the transition zone between them characterized by small diameters (central-left panel) and low rainfall rates. Unfortunately, the rainfall structure is not preserved above the freezing level. A sharp discontinuity of precipitation rates through the melting zone is particularly evident for high intensities, where approximately a fourfold increase is observed. Such results are at odds with melting models, where constant mass flux is usually assumed (e.g. Szyrmer and Zawadzki, 1999; Zawadzki et al., 2005; Matrosov, 2008) or the precipitation rate from snow to rain changes gradually (e.g. Heymsfield et al., 2018a; Mróz et al., 2021).

## 3.2   Statistical relationship between ice and rain parameters

In order to have a better insight into the relation between PSD parameter within ice and rain, a histogram of the retrieved values of $D_m$ and $N_w$ below and above the BB is shown in Figure 4. To mitigate biases arising from errors in detecting the melting layer extent, we define the data 500 m above the detected bright band top as the region "above the melting layer." This approach assists in screening cases where the bright band top altitude is uncertain and the top of the melting layer coincides with a region of sharp reflectivity increase.

The sensitivity of the DPR system combined with the assumptions of the algorithm limits the lowest detectable size at 0.5 mm (Figure 4 a). Because of the ambiguity in the inversion process for small DFRs (Seto et al., 2013), the PSDs with $D_m$ smaller than 1 mm are extremely rare. On the other hand, there is a hard limit of 3 mm for a maximal characteristic size as it was already observed by Gatlin et al. (2020). The retrieval suggests an increase of approximately 18% in the $D_m$ while particles are falling through the melting zone. The growth can be partially caused by the vapour deposition, which contributes to the increment of $\sim 6\%$ in mass (Wexler, 1955), i.e., only 2% in terms of the size. The remaining 16% of the "observed" growth should be attributed to aggregation, or it may result from the algorithm underestimation of the assumed ice density/size or overestimation of rain diameters as it has been reported by Petersen et al. (2018) and D'Adderio et al. (2018). Surprisingly, the statistical relation between retrieved characteristic sizes above and below the BB contradicts the findings of Mitra et al. (1990), who observed that snowflake breakup during melting is not uncommon, especially for strongly asymmetric particles. The recent melting simulations by Leinonen and Lerber (2018) confirm that unrimed particles are prone to breakup, indicating larger particles (in terms of melted diameter) should be expected above the BB, not below. However, these aforementioned studies did not consider the possibility of particle collision and coalescence within the melting zone.

The retrieved intercept parameter values are closely clustered around $10^{3.5}$ mm$^{-1}$ m$^{-3}$, with negligible variations among different water phases (refer to Figure 4 b). This observation elucidates the rapid changes in mass flux during melting in the DPR product. In order to maintain the precipitation rate, the concentration of slowly falling snowflakes must be considerably higher than that of rapidly falling raindrops, as posited by the mass flux conservation principle underlying formula (3). The potential underestimation of ice concentration aligns with recent findings by Chase et al. (2021), who assessed DPR retrieval using airborne radar data and collocated in-situ probes, as well as the study by Mroz et al. (2021b), who demonstrated a large underestimation of snowfall rates over the continental US in comparison with ground-based radar products. Similar problem

with the snowfall deficiency in the 2A.DPR product was also reported by Skofronick-Jackson et al. (2019) and Casella et al. (2017). The underestimation of ice particle concentration implies a severe discontinuity of the water mass flux through the melting zone. The panel (c) of the Figure 4 shows that the snowfall rate above the BB is roughly one third of the rain rate below the BB.

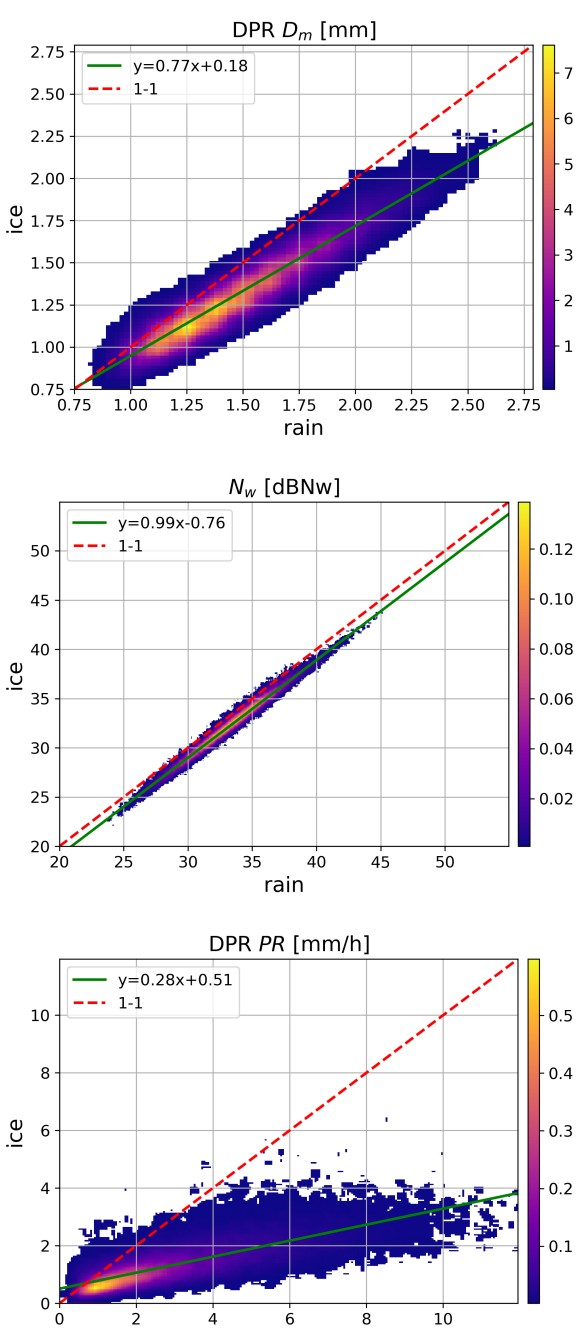

**Figure 4.** As in Fig. 2 but for the retrieved PSD parameters in the Level-2 V06 DPR product.

In-depth analysis of the modeled Ka-band reflectivity above the bright band (BB) revealed that the radar signal corrected for attenuation is, on average, smaller than the actually measured reflectivity, as shown in Figure 5. While some differences

between the two are expected due to random noise in the measured reflectivity, systematic underestimation indicates non-physical negative attenuation or issues in the radar simulator's ability to fit the measurements. These discrepancies can have several consequences for the retrieval process with the overestimation of the characteristic size of ice particles being the most obvious.

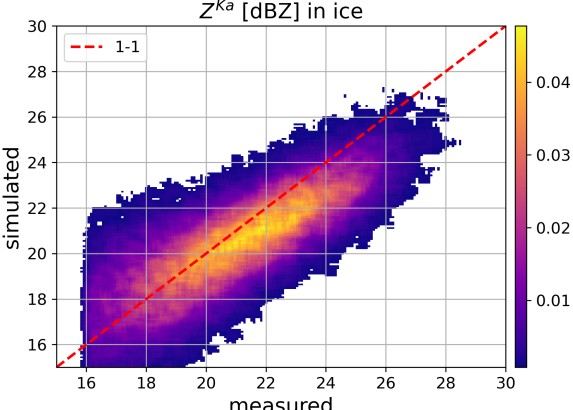

**Figure 5.** A PDF of the measured and the corrected for attenuation Ka-band reflectivity above the melting region in the DPR V06 precipitation product.

We conducted a similar analysis with another official GPM product, specifically the 2B.GPM.DPRGMI.CORRA. This algorithm integrates DPR data with passive measurements from the GPM Microwave Imager (GMI). We evaluated the same version of this product over a consistent time span of 5 years. The findings indicated that this product is also subject to the same issue—specifically, the precipitation rate above the melting layer is only one-third of the rainfall rate below the melting zone. The regression line $PR(ice) = 0.29PR(rain) + 0.89$ best describes the mass flux change between the phases. Similar to the radar-only product, the melted equivalent size of precipitation is well preserved through the melting zone $(D_m(ice) = 0.83D_m(rain) + 0.14)$.

The striking similarities between the two official DPR products were already observed by Mroz et al. (2021b), who noted that both products are affected by a similar underestimation of snowfall rates over the continental US. For brevity and given the similarities to the corresponding figures for the DPR product, the histograms illustrating these results are not included in this paper.

## 4  OE retrieval development

The retrieval method proposed here aims to loosely enforce the continuity of the melted diameter and the precipitation rate as particles transition through the melting zone. It follows the DPR approach with the use of a gamma PSD (4); however no assumption on the shape parameter $\mu$ is made. Following, Delanoë et al. (2005) we parametrize the PSD using a concept of

moment normalization, i.e.:

$$M_p = \int\limits_0^{D_{\max}} D^p N(D)\,\mathrm{d}D, \tag{5}$$

where $p$ is the moment, $D$ is the liquid sphere equivalent diameter and $D_{\max}$ is the diameter of the largest particle. With this approach, three parameters of the gamma model ($N_w$, $D_m$, $\mu$) have an equivalent representation in terms of the physical quantities such as (PR, $D_m$, $\sigma_m$) defined as

$$\mathrm{PR} = \int\limits_0^{D_{\max}} N(D)v(D)m(D)\,\mathrm{d}D, \tag{6}$$

$$D_m = M_3^{-1} M_4, \tag{7}$$

$$\sigma_m^2 = M_3^{-1} \int\limits_0^{D_{\max}} D^3(D - D_m)^2 N(D)\,\mathrm{d}D, \tag{8}$$

where $m$ and $v$ denote the mass and velocity of particles, respectively. $D_m$ represents the mass-weighted mean diameter of the PSD, while $\sigma_m$ indicates the mass-weighted standard deviation, often referred to as the width of the PSD. PR denotes the precipitation rate, which serves as the scaling parameter for particle concentration:

$$N(D; \mathrm{PR}, D_m, \sigma_m) = \mathrm{PR} \times f(D; D_m, \sigma_m). \tag{9}$$

The function $f(D; D_m, \sigma_m)$ describes the shape of the PSD at a unit precipitation rate for a given characteristic size and width of the PSD. The aim of the DPR OE retrieval algorithm is to retrieve all 3 mentioned descriptors of the PSD, therefore, the vector of the retrieved parameters has the following form:

$$\begin{aligned}
\mathbf{x}^{\mathrm{ph}} = \big[ &\log_{10} \mathrm{PR}^1,\ \log_{10} \mathrm{PR}^2, \ldots \log_{10} \mathrm{PR}^N, \\
&\log_{10} D_m^1,\ \log_{10} D_m^2, \ldots \log_{10} D_m^N, \\
&\log_{10} \sigma_m^1,\ \log_{10} \sigma_m^2, \ldots \log_{10} \sigma_m^N \big]^T
\end{aligned} \tag{10}$$

where $N$ is the total number of DPR range gates above the Ku-radar sensitivity threshold, while the superscript "ph" emphasizes that these are physical quantities.

## 4.1 A-priori assumptions

Our a-priori assumptions are specifically designed to align with the specifications of the DPR system, encompassing factors such as radar volume size and system sensitivity. To achieve this, we employ a "scale-up" approach, where we derive statistics on precipitation properties from ground-based polarimetric radars and average them to match the resolution of the DPR through collocated match-up volumes (Gatlin et al., 2020). This involves utilizing a network of S-and C-band radars that are part of the GPM ground validation (GV) program. It is important to note that ground-based radars offer finer horizontal resolution

compared to the space-borne system, thanks to their shorter distance to the meteorological targets. This enhanced resolution contributes to more detailed observations and reduces the problem of precipitation heterogeneity within the radar volume.

The collocated match-ups of radar volumes facilitate effective cross-calibration between space-borne and ground-based sys-
tems. Additionally, state-of-the-art polarimetric microphysical retrieval techniques are employed to estimate the characteristic rain size (Tokay et al., 2020) and precipitation rate (Cifelli et al., 2011) at fine scales, which are subsequently averaged to match the footprint size of the DPR. The polarimetric retrieval methods relies on radar simulations of disdrometer measurements and are validated with the ground based rain gauges. This allows the GV dataset to bridge the gap between very distinct observation scales of disdrometers and the DPR.

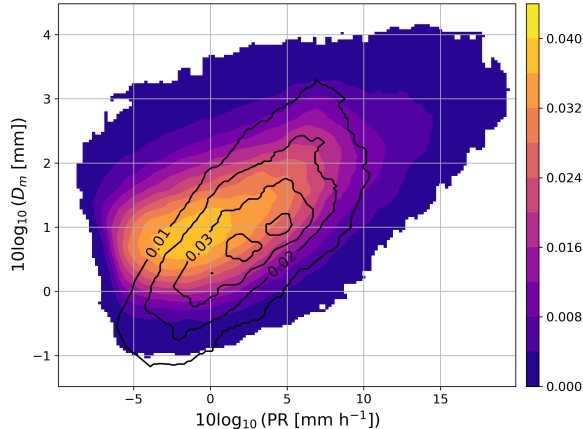

**Figure 6.** Joint PDF of $D_m$ and PR at GPM DPR resolution obtained through polarimetric radar retrieval. The retrieval is based on radar simulations of in-situ DSD observations. In-situ disdrometer data shown as a black contour line for comparison. Both PDFs include only samples where radar reflectivity at the S-band exceeds 15 dBZ.

Figure 6 illustrates the difference between the measured DSD characteristics at different scales. The black contour line represents the joint PDF of $D_m$ and PR, as observed by 2DVD disdrometers deployed for the GPM ground validation program (Dolan et al., 2018). The colour shading represents the data at the DPR resolution from the GV program data. Notably, the large-scale measurements cover a broader domain compared to the in-situ data, despite the expected smearing-out effect in larger volumes. Additionally, the bull-eye of the distribution is shifted to the left by 5 dB (equivalent to a factor of 3 difference),
indicating that at the DPR resolution, smaller precipitation rates are recorded for the same $D_m$.

In our analysis, we explore the statistical relations between the microphysical properties of rain derived from the polarimetric radar retrieval. These relationships are expressed in terms of the covariance matrix, $\mathbf{S_b^{ph}}$, and the corresponding sample mean,

$\mathbf{x}_\mathbf{b}^{\mathrm{ph}}$, of the parameters under consideration:

$$\mathbf{S}_\mathbf{b}^{\mathrm{ph}} = \begin{pmatrix} 22.441 & 2.307 & 3.229 \\ 2.307 & 0.717 & 1.067 \\ 3.229 & 1.067 & 2.094 \end{pmatrix}, \tag{11}$$

$$\mathbf{x}_\mathbf{b}^{\mathrm{ph}} = \begin{bmatrix} 2.493, & 1.262, & -3.044 \end{bmatrix}^T, \tag{12}$$

It is important to note that $\mathbf{x}_\mathbf{b}^{\mathrm{ph}}$ and $\mathbf{S}_\mathbf{b}^{\mathrm{ph}}$ represent a priori assumptions made independently for each altitude level. To determine the contribution of the divergence of $\mathbf{x}^{\mathrm{ph}}$ from the statistical mean to the cost function, as introduced in equation (19), we aggregate the components across all altitude levels.

To mitigate the influence of outliers in the dataset, a robust methodology known as the Minimum Covariance Determinant
(Butler et al., 1993) was employed for the analysis. It is important to note that $10\log_{10}\sigma_m$ is not directly retrieved by the polarimetric product, and therefore disdrometer measurements were utilized to derive statistical information regarding this parameter. To improve the convergence rate of the OE algorithm, the dataset underwent Principal Component Analysis (PCA) to diagonalize the matrix $\mathbf{S}_\mathbf{b}^{\mathrm{ph}}$. Prior to PCA, data normalization was performed using the formula $x \mapsto (x - \mathrm{mean}(x))/\mathrm{std}(x)$. The resulting principal components are as follows:

$$\mathrm{PC}_1 = \begin{bmatrix} 0.497, & 0.625, & 0.602 \end{bmatrix}^T, \tag{13}$$

$$\mathrm{PC}_2 = \begin{bmatrix} 0.858, & -0.252, & -0.447 \end{bmatrix}^T, \tag{14}$$

$$\mathrm{PC}_3 = \begin{bmatrix} 0.128, & -0.739, & 0.662 \end{bmatrix}^T, \tag{15}$$

Each principal component explains 76%, 20%, and 4% of the variability in the dataset, with corresponding explained variances of 2.29, 0.59, and 0.12, respectively. These principal components represent orthogonal directions within the microphysi-
cal properties space. The first principal component signifies the most probable direction of change when only one measurement is available, while the second principal component captures the correction introduced by an additional measurement.

## 4.2 The forward model and the measurements vector

To estimate microphysical properties of precipitation, it is crucial to understand their relationship with radar measurements. This requires a radar simulator that maps the physical characteristics of precipitation to radar observables. In our specific case,
the forward model is responsible for mapping the microphysical parameters (PR, $D_m$, $\sigma_m$) to radar reflectivity at both the Ka- and Ku-band frequencies.

The equivalent reflectivity factor for a radar operating at the wavelength $\lambda$ [m] is given by:

$$Z_e^\lambda(\mathrm{PR}, D_m, \sigma_m) = \frac{\lambda^4}{\pi^5 |K_w^\lambda|^2} \int_0^\infty \sigma_b(D,\lambda) N(D) \ \mathrm{d}D$$

$$= \frac{\lambda^4 \, \mathrm{PR}}{\pi^5 |K_w^\lambda|^2} \int_0^\infty \sigma_b(D,\lambda) f(D; D_m, \sigma_m) \ \mathrm{d}D \tag{16}$$

where $\sigma_b$ is the backscattering cross-section of a particle [m$^2$] and $K_w^\lambda$ is the dielectric factor of liquid water at a reference temperature and frequency and is assumed to be equal to 0.9255 and 0.8989 for Ku- and Ka-band, respectively, as these values are used in the DPR Level 2 processing chain (Liao and Meneghini, 2022). The reflectivity is usually expressed in mm$^6$ m$^{-3}$ or, due to its high variability, in the logarithmic units of dBZ. Similarly, specific attenuation $k$ [dB km$^{-1}$] can be computed as

$$k^\lambda(\mathrm{PR}, D_m, \sigma_m) = \frac{10}{\ln 10} \int_0^\infty \sigma_e(D) N(D) \mathrm{d}D$$

$$= \frac{10 \, \mathrm{PR}}{\ln 10} \int_0^\infty \sigma_e(D) f(D; D_m, \sigma_m) \mathrm{d}D, \tag{17}$$

and the measured reflectivity at distance $r$ is given by eq. 91).

The path integrated attenuation (PIA), represented by the integral $\int_0^r k(s) \ \mathrm{d}s$, provides valuable information about precipitation properties. However, estimating this quantity is challenging, particularly over land, due to surface variability. Nonetheless, a study by Meneghini et al. (2015) demonstrated that the difference in path integrated attenuation between the Ka- and Ku-bands, denoted as $\delta$PIA, is less affected by surface variability and thus exhibits smaller estimation errors. Based on this finding, we have made the decision to include $\delta$PIA in the measurement vector, rather than individual PIA estimates.

The backscattering and extinction cross-sections of ice particles are modelled with the database of rimed snowflakes of Mroz and Leinonen (2023). The scattering characteristics of the simulated particles were calculated using the discrete dipole approximation (DDA), which is known to provide more realistic radar measurements compared to commonly used soft spheroids (e.g. Kneifel et al., 2015; Kulie et al., 2014; Leinonen et al., 2012). The database encompasses a wide range of ice densities and sizes, making it well-suited for our study. The bulk density of ice particles within the radar volume is characterized by the degree of riming, which is parametrized by a prefactor term ($\alpha$) in the mass-size relationship of the form $m[\mathrm{kg}] = \alpha(D[\mathrm{m}])^2$. The database covers ice hydrometeors ranging from unrimed aggregates ($\alpha = 0.01$) to graupel-like particles ($\alpha = 0.5$). For a more detailed description of this conceptual model, please refer to Mroz et al. (2021a).

The scattering properties of liquid particles are computed with the T-matrix approximation, assuming the axial-ratio formula of Brandes et al. (2005). We do not explicitly model radar echoes within the melting region due to the complex scattering signatures and associated uncertainties involved. However, we account for the impact of the bright band on the measurements within the liquid phase below by simulating its extinction. Drawing from the findings of Matrosov (2008), we approximate the attenuation of the melting layer using a power-law formula based on the precipitation rate:

$$A_{\mathrm{ML}}[\mathrm{dB}] = \gamma_{\mathrm{ML}} \left( \mathrm{PR}[\mathrm{mm \ h}^{-1}] \right)^{\delta_{\mathrm{ML}}}. \tag{18}$$

Here, $\gamma_{\mathrm{ML}}$ and $\delta_{\mathrm{ML}}$ are wavelength-specific parameters. For the Ka-band, their values are 0.66 and 1.1, respectively. In the case of Ku-band simulations, we adopt the values obtained from X-band simulations, specifically 0.048 for $\gamma_{\mathrm{ML}}$ and 1.05 for $\delta_{\mathrm{ML}}$. Although we acknowledge that X-band attenuation is likely to be smaller than that of Ku-band, we use it solely as a soft constraint or a priori value. The final attenuation estimate is subsequently refined during the OE iterations. In the study of Li and Moisseev (2019), it was suggested that synthetic simulations by Matrosov tend to overestimate attenuation for snowfall rates exceeding $2.5 \ \mathrm{mm;h^{-1}}$. However, their study was limited to radar measurements exhibiting clear signatures of supercooled clouds above the freezing level. This limitation implies that the study was restricted to rimmed particles only. To accommodate potential variations in the melting layer attenuation estimates, we operate under the assumption that they are subject to a factor of 2 uncertainty (see the next section). It is important to note that this approach focuses solely on simulating the extinction of the bright band and its influence on the measurements underneath.

### 4.3 Optimal estimation concept

The inversion method presented here is based on the optimal estimation framework (Rodgers, 2000), that aims at minimizing the cost function:

$$CF(\mathbf{x}) = (F(\mathbf{x}) - \mathbf{y})^T \mathbf{S_m^{-1}} (F(\mathbf{x}) - \mathbf{y})$$

$$+ (\mathbf{x} - \mathbf{x_b})^T \mathbf{S_b^{-1}} (\mathbf{x} - \mathbf{x_b}). \quad (19)$$

In our case, the vector $\mathbf{x}$ consists of the PSD parameters, namely PR, $D_m$, and $\sigma_m$ (as shown in equation 10), for both the ice and rain phases. Additionally, within the solid phase, the parameter $\alpha$, which quantifies ice density, is also retrieved. Notably, all microphysical parameters are estimated in the logarithmic space by employing the $x \mapsto 10 \log_{10} x$ transformation of the physical quantities. This approach serves two purposes: preventing unphysical retrievals and promoting linearity in the forward model. It is assumed that $10 \log_{10} \alpha$ changes linearly from the cloud top, where it is equal to -20, to the melting layer. Furthermore, the vector of unknowns includes an additional parameter, denoted as $\mathrm{BB_{ext}}$, which characterizes the correction of the attenuation formula (18) within the BB region. This parameter accounts for uncertainties in the parametrization by multiplying the attenuation estimates at both the DPR frequencies. In summary,

$$\mathbf{x} = \left[ \mathbf{x}^{\mathrm{ph}}; \ 10 \log_{10} \alpha, \ 10 \log_{10} \mathrm{BB}_{\mathrm{ext}}^{Ku}, \right.$$

$$\left. 10 \log_{10} \left( \mathrm{BB}_{\mathrm{ext}}^{Ka} - \mathrm{BB}_{\mathrm{ext}}^{Ku} \right) \right].$$

The covariance matrix terms corresponding to the physical quantities are provided by equation (11). Similarly, the a-priori value at a given altitude level is given by equation (12). The a-priori estimate for $10 \log_{10} \alpha$ is determined based on the radar measurements. We obtain the expected values of precipitation rate, size, and shape parameters by aligning the Ku reflectivity measurements below the melting level with the reflectivity simulations along the direction given by the first principal component (equation 13). The other two principal components are set to 0. Once this one-dimensional search yields the best match, we calculate radar reflectivity in ice for different values of $\alpha$, assuming that microphysical parameters remain constant within the

melting zone. The value of alpha that provides the best fit to the measurements, weighted by their corresponding uncertainties, is selected as the a-priori estimate. We assume that the standard deviation of this estimate is equal to 3 dB. The uncertainties of $10\log_{10} \mathrm{BB}_{\mathrm{ext}}^{Ku}$ and $10\log 10 \left(\mathrm{BB}_{\mathrm{ext}}^{Ka} - \mathrm{BB}_{\mathrm{ext}}^{Ku}\right)$ are assumed to be 3 dB and 4 dB, respectively. A priori estimates of $\mathrm{BB}_{\mathrm{ext}}^{Ku}$ and $\mathrm{BB}_{\mathrm{ext}}^{Ku}$ are provided by equation (18).

The function $F$ is the aforementioned forward model that transforms physical quantities into attenuated reflectivity at both bands and the differential PIA. The vector $\mathbf{y}$ consists of the measured values of Z and the differential PIA that are corrected for attenuation by non-precipitating particles and atmospheric gases (Kubota et al., 2020). The matrix $\mathbf{S_m^{-1}}$ is an inverse of the measurement error covariance matrix, and it is used as a weighting factor for individual observables. We assume that $\mathbf{S_m}$ is diagonal and the uncertainty of the DPR reflectivity at both channels is estimated by the formula given by Hogan et al. (2005), which gives 0.5 dB uncertainty for high signal-to-noise ratio data. The uncertainty of the estimated differential PIA is assumed to be five times the uncertainty of the PIA at the Ku-band in the DPR Level 2 product, following the assumption that $PIA^{Ka} = 6PIA^{Ku}$.

To fully utilize the potential of Principal Component Analysis (PCA), we transform the estimation problem of physical quantities into the space of principal components. This linear transformation ensures that both approaches are equivalent. The transformation can be expressed by the following equation:

$$\mathbf{x^{ph}} = (\mathbf{P} \times \mathbf{a}) \odot \sqrt{\mathrm{diag}(\mathbf{S_b^{ph}})} + \mathbf{x_b^{ph}}, \tag{20}$$

where P is a matrix whose columns are the principal components, i.e., $\mathrm{P} = (\mathrm{PC}_1; \mathrm{PC}_2; \mathrm{PC}_3)$, $\mathbf{a}$ is a vector in the principal component basis, $\odot$ denotes element wise multiplication, and $\times$ represents matrix multiplication.

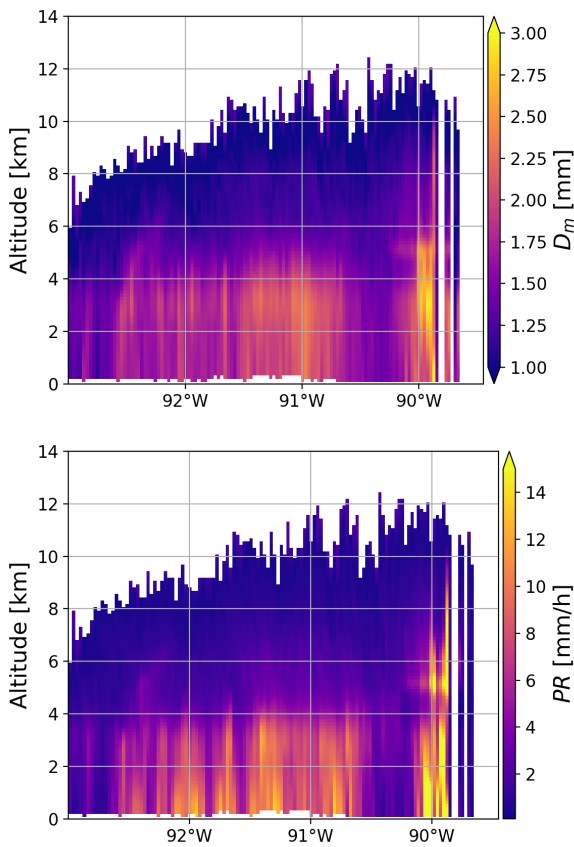

**Figure 7.** Mean mass weighted melted diameter ($D_m$) and precipitation rate ($PR$) as estimated by the optimal estimation algorithm for the storm depicted in Fig. 1.

An example of the OE retrieval is presented in Fig. 7. The rain component aligns with the overall structure observed in the DPR product, featuring distinct elements of a squall line system: a convective core, a transition zone, and stratiform precipitation, viewed from east to west. However, the range of retrieved parameters is smaller compared to the DPR algorithm, even within stratiform rain profiles. This discrepancy is mainly caused by differences in the differential attenuation fitting between the two algorithms. The DPR product tends to overestimate this parameter as it tries to compensate for the non-uniform beam filling effect, while our product does not incorporate this correction. It must acknowledged that the reliability of the OE product is questionable in convective precipitation, given its design for stratiform precipitation. Consequently, the presence of artifacts at approximately 5 km altitude (a peak in $D_m$ and $PR$) within the convective core should not come as a surprise.

In the ice phase, the OE algorithm tends to yield a larger precipitation rate than the DPR product. Both the $PR$ and $D_m$ reach their peaks above the melting zone and remain relatively constant in rain; that is, the majority of the growth occurs within the ice phase. Moreover, compared to the DPR product, the precipitation structure in the ice phase is not affected by

strong discontinuities during the transition from one profile to the other, although no constraints on horizontal variability were imposed.

## 4.4 Similarities and differences with the DPR product

In spite of the distinct mechanics employed by our algorithm, specifically our reliance on the optimal estimation framework, and the iterative nature of the DPR product, which primarily aims at fitting the measurements, there exist notable similarities between these two approaches. For instance, the utilization of principal components in our method shares an underlying idea with the $PR$-$D_m$ relationship. The principal components determine orthogonal directions within the space of microphysical parameters while providing insights into which component is most likely to change. The first principal component, for instance, represents the direction that undergoes the most significant changes as it is characterized by the largest variance, by definition. Variations along this principal component can be likened to imposing the $PR$-$D_m$ relationship, a step analogous to the approach adopted in the DPR product. A noteworthy similarity arises when altering the second principal component; this modification influences the $PR$-$D_m$ relationship, akin to the $\epsilon$-adjustment implemented in the DPR product. Despite these similarities, our approach offers a distinct advantage – a priori knowledge regarding the natural variability of these relationships, quantified by their respective standard deviations. This insight allows for a more nuanced understanding of how these relationships may vary in real-world scenarios.

Another notable similarity between our algorithm and the DPR product lies in the approach to assimilating the measured reflectivity. Traditionally, the measured reflectivity is corrected for attenuation prior to microphysical retrievals (e.g. Vulpiani et al., 2006). Both our and the DPR algorithm adopt this step solely to obtain the initial guess. Subsequently, an iterative procedure is initiated, and the distribution of microphysical parameters within the column is modified to align with the measured reflectivity. Both algorithms employ a top-down approach, estimating attenuation caused by various hydrometeors from scattering tables. The attenuation accumulates along the propagation path until reaching the surface. The total PIA estimate serves as a crucial constraint for both algorithms, ensuring the stability of the iterative process. The difference lies in the modeling of the melting layer. In our approach, we solely estimate the attenuation caused by melting particles using the parametrization of Matrosov (2008). In contrast, the DPR product simulates the melting particles and their associated scattering properties within the melting zone. This simulation yields reflectivity and hydrometeor properties profiles within the melting zone. In our case, hydrometeor properties are obtained solely through continuity, and no measurements are simulated within the melting zone.

It is essential to highlight a nuanced difference in our algorithm compared to the DPR product. Our algorithm is designed to simultaneously fit the measured reflectivity at Ku and Ka bands, alongside the differential PIA estimate. Conversely, the DPR product appears to prioritize fitting the Ku-band reflectivity. This prioritization is justified due to challenges in simultaneously fitting both channels under non-uniform conditions (Mroz et al., 2018). Notably, our algorithm is tailored for stratiform rain scenarios, where such conditions are minimized, while the official DPR product is designed to be a versatile one-for-all approach.

The primary distinction between the two algorithms centers on how ice is modeled. Our approach employs the simulation of realistic ice hydrometeors, complemented by discrete dipole simulations of scattering properties. In contrast, the DPR

product adopts "soft-spheres" simulations, representing ice particles as a uniform mixture of air and ice. In this case, scattering simulations can be approximated using Mie theory. However, it's essential to emphasize that the primary difference in our approaches is not the shape of the particles or the scattering simulation methodology. What sets our algorithm apart is the capacity of the OE algorithm to accommodate changes in the density of ice particles, while the DPR product maintains a fixed density of spherical air-ice mixtures at $0.1\,\mathrm{g\,cm^{-3}}$. This unique flexibility allows the OE algorithm to search for solutions that ensure continuity in the water mass flux through the melting zone. Importantly, this coherence in the fluxes is achieved without sacrificing the continuity of the melted equivalent size, and it is obtained with the radar measurements matching. The ability to adjust ice particle density provides a crucial advantage, enabling our algorithm to navigate to physically consistent solution more effectively.

## 5   Demonstrating Criteria Compliance: Algorithm Evaluation

The Optimal Estimation algorithm underwent testing if it achieves its core retrieval objectives, which include ensuring continuity in precipitation properties and evaluating the accuracy of simulated reflectivity against actual measurements. A summary of this testing is presented in Figure 8. Notably, our algorithm yields a narrower range of retrieved raindrop sizes compared to the DPR product, and when examining the regression line, it suggests a more substantial increase in precipitation size during the melting phase. It's worth noting, however, that the observed increase consistently remains below 30%, whereas the DPR product showed an approximate 15% increase.

Notably, our product improves the coherence between precipitation rates above and below the melting zone. Similar to the DPR product, there is an increase in water mass flux during the transition from ice to rain, but in our case, this increase is more subdued. Upon examining the regression lines, the expected snowfall rate above the melting zone, with 10 mm/h of rainfall underneath, equals 6 mm/h for our product and 3 mm/h for the DPR product.

Overall, the mean fractional bias (MFB), defined as:

$$MFB = \exp\left[\frac{1}{N}\sum_{i=1}^{N}\left(\ln PR_{rain}^{i} - \ln PR_{ice}^{i}\right)\right] - 1, \tag{21}$$

is equal to 4% and 106% for the OE and the DPR algorithms, respectively. Using DPR precipitation rates in rain as the reference for our product, the snowfall rate is on average 30% smaller than the rainfall rate underneath.

In the validation study conducted by Chase et al. (2022), it was demonstrated that the Neural Network snowfall algorithm (Chase et al., 2021), designed specifically for the DPR, exhibits improved agreement between snow and ice phase precipitation rates compared to the DPR product. Their algorithm almost perfectly aligns with the mass flux between the phases, showing only a 2% difference. This underscores the remarkable success achievable with artificial intelligence algorithms when trained using the right database. However, it's crucial to note that its accuracy is contingent on the precision of the attenuation correction, particularly at the Ka-band, which is more susceptible, especially in heavy precipitation conditions.

While our algorithm doesn't achieve a perfect precipitation rate match between different water phases, it brings about noticeable improvement compared to the DPR product. The algorithm's development is still in its early stages, and ongoing

adjustments in the code are expected to reduce this discrepancy. Preliminary tests have been conducted, revealing that the continuity of the mass flux through the melting zone can be enhanced at the expense of the continuity of the melted equivalent size. However, it remains debatable which of these two properties exhibits less variability within the melting zone, and this should be a topic for future studies. Furthermore, it's important to acknowledge that some changes in the precipitation rate may still occur within the melting zone, resulting in an anticipated growth in precipitation rate. The extent of this change depends

on environmental conditions such as relative humidity and temperature profiles (Heymsfield et al., 2018b).

Upon evaluating the Ka-band reflectivity simulations in the ice phase region, we observe that our algorithm demonstrates improved accuracy when compared to the official product (as shown in Fig. 5). Our simulations exhibit an absence of bias. In fact, the majority of our simulated values closely align with the measured values, typically falling within a range of just 1 dB. Having said that, it's important to note that the reflectivity values below the 20 dBZ threshold exhibit higher uncertainties. In

the current version of our algorithm, we consider these measurements as marginally reliable, and accordingly, we do not expect the algorithm to fit them well.

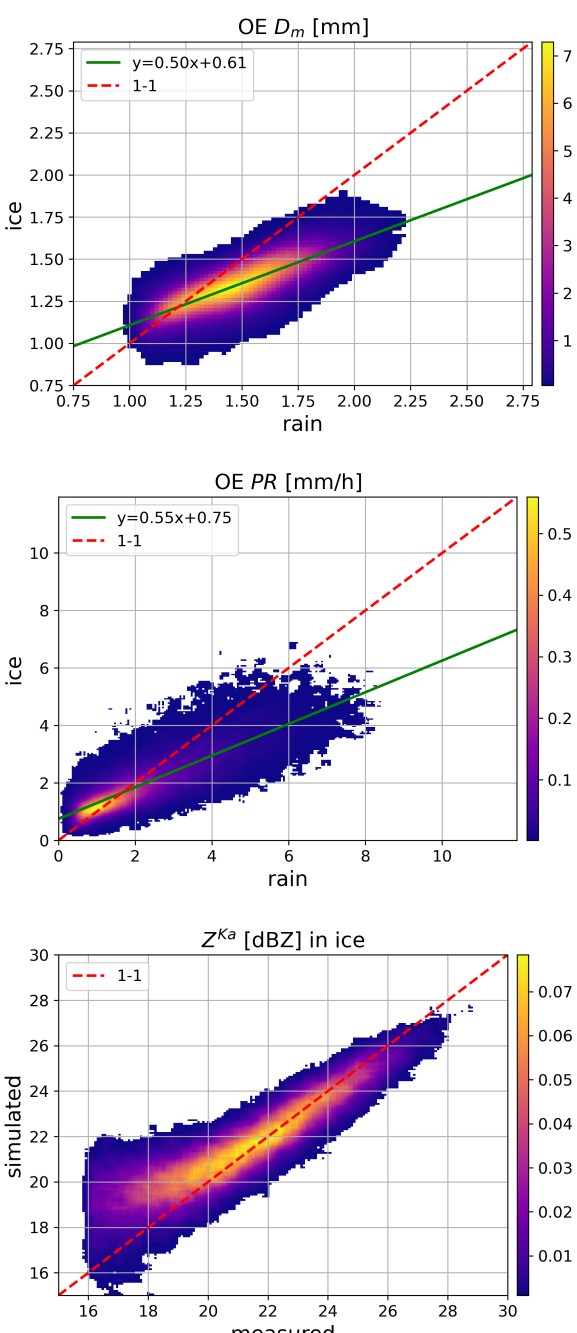

**Figure 8.** The top two panels display joint PDFs for OE retrieval results both above and below the melting zone. A green line signifies a linear fit through the most frequently observed values for the variable $x$. The top panel depicts $D_m$, the middle panel shows $PR$, and the bottom panel illustrates PDFs for measured and simulated Ka-band reflectivity values, differentiating between regions above and below the melting zone.

## 6 Performance assessment in rain

The validation of the OE algorithm was exclusively conducted within the rainy portion of the radar profiles. This might appear surprising, given the anticipated improvement in algorithm quality compared to the DPR product above the freezing level. However, this approach is expedient due to the limited availability of DPR under-flights within snow during stratiform precipitation events. To the best of our knowledge, only one flight was conducted throughout the entire OLYMPEX campaign, and this singular event was utilized in the study by Chase et al. (2021). In their study, only a qualitative assessment of the product was conducted, refraining from direct comparisons due to disparities in sampling time during in-situ flights and significant differences in sampling volume. The discrepancy in sampling time arises from the high ground track speed of the satellite ($7 \, \text{km} \, \text{s}^{-1}$) compared to approximately $600 \, \text{km} \, \text{h}^{-1}$ of an in-situ aircraft. Consequently, within a 10-minute window, only 20 validation points are collected. This raises a critical question about the representativeness of the sample and the robustness of potential statistical comparisons. Moreover, in-situ sampling may be inadequate to sufficiently represent the entire radar volume, given its proximity to a one-dimensional cut through a $5 \times 5 \times 0.25 \, \text{km}^3$ volume. The impact of this difference in sampling volume could potentially be mitigated with the collection of large statistics, as discrepancies in the sampling volumes would result in random noise only. However, as pointed out earlier, collecting these statistics is impractical due to the limited number of validation points per flight, making such an effort very expensive. Chase et al. (2021) overcame this issue by utilizing airborne radar data at finer horizontal and vertical resolutions for more robust statistics. While we acknowledge their efforts, it's crucial to note that airborne data differ significantly from spaceborne measurements. Airborne data exhibit superior sensitivity, resolution, and reduced signal fluctuations. Additionally, they are less affected by non-uniform beam filling effects compared to satellite measurements. The validation presented here served as a sanity check, aiming to assess whether a physically consistent retrieval could be achieved without compromising the integrity of the DPR rainfall product.

We employed the state-of-the-art polarimetric retrieval method, relying on ground-based radar measurements, as our reference dataset. Our choice of this validation strategy is driven by the fact that a single radar network covers a much larger area compared to the collective measurement coverage of all rain gauges worldwide. Furthermore, due to the short correlation lengths of precipitation patterns, rain gauges cannot accurately represent instantaneous precipitation rates.

For our validation, we chose a dataset from Gatlin et al. (2020) specifically designed for GPM validation. This dataset offers quality-controlled dual-polarimetric radar moments and corresponding polarimetric estimates of DSD parameters, such as $D_m$ and $PR$, obtained from approximately 100 radar systems worldwide. The dataset is based on the concept of matching space-borne and ground-based radar volumes that was originally introduced by Schwaller and Morris (2011). High horizontal resolution DSD moments data (with a resolution of 250 meters and 1-degree in range and azimuth, respectively) are averaged to generate GPM DPR footprint matches. Similarly, an averaging of the DPR data, known for its high vertical resolution, is applied to align with the ground-based radar volumes. The validation is performed with all the collocated radar volumes that are not contaminated by the ground clutter or melting layer signatures.

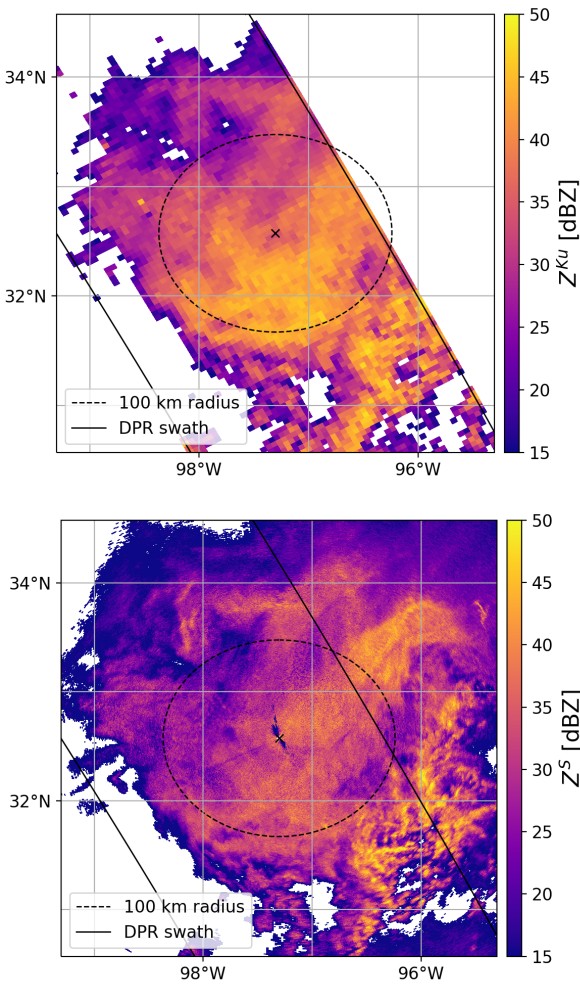

**Figure 9.** Top Panel: Ku-band radar reflectivity at an altitude of 3 km ASL, as observed by the DPR. Bottom Panel: S-band reflectivity obtained from the lowest elevation scan (0.5°) conducted by the ground-based radar situated in Dallas, US (KFWS). These measurements are associated with DPR orbit number 11525, which passed over the storm being observed at around 10:10 UTC on 23^rd July, 2018. The black continuous lines indicate the edges of the DPR swath while the dashed line shows 100 km radius from the ground based station.

A comparison between the horizontal resolution of the DPR system and that of a US ground-based radar is illustrated in
Fig. 9. The correlation between the two radar fields is noteworthy. Nevertheless, owing to its higher horizontal resolution, the ground-based data reveals much finer structures, particularly noticeable in the convective cells situated in the southeastern part of the precipitation area. It's important to highlight that the ground-based radar exhibits greater sensitivity, enabling the validation of DPR precipitation retrieval across the full range of detectable signals. Additionally, at short ranges, where the radar beam volumes are close to the ground, the issue of precipitation overshooting is minimized.

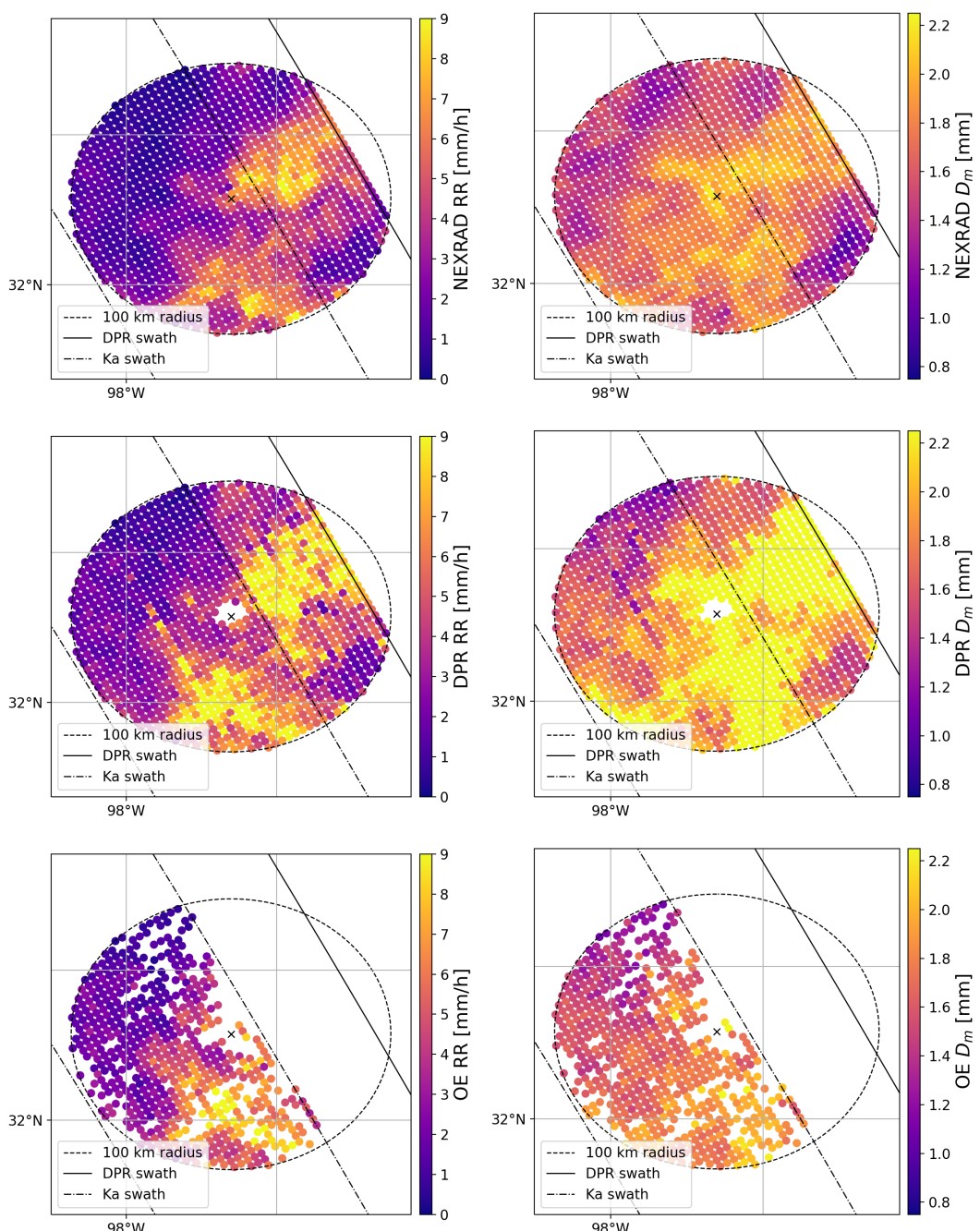

**Figure 10.** Microphysical precipitation retrievals within the matched radar volumes at the lowest available altitude. The data is confined within a 100 km radius from the ground-based station to preserve high vertical resolution. The dashed-dotted line indicates the edge of the inner swath where the Ku and Ka band radar perform collocated measurements. The continuous line shows the edge of Ku band swath. The left column shows precipitation rate while the right column precipitation size, $D_m$. Top Panel: Dual-polarization retrieval from the ground-based station; Middle Panel: DPR rain rate product; Bottom Panel: OE-derived precipitation rate.

The left column in Fig. 10 illustrates rain rate retrievals within a 100 km radius from the ground-based station, focusing on the radar-matched volumes at the lowest available altitude. The dual-polarization rainfall product is derived using the methodology presented in Cifelli et al. (2011). A visual examination of the DPR precipitation product for this event indicates an overestimation of high rainfall rates. However, this overestimation is less pronounced within the matched scan (MS) swath, where Ku and Ka-band radars closely collocate in both space and time. This observation highlights the superior performance of the dual-frequency algorithm compared to the Ku-only product. The OE algorithm was exclusively applied to columns within the MS swath, where the bright band was distinctly detected by the DPR algorithm. This results in gaps in the image compared to the DPR product. Nonetheless, despite these challenges, the storm structure closely resembles that of the polarimetric product. It is important to note that we cannot assess its performance within the strongest echoes, as they were classified as convective profiles and, consequently, were not processed by our algorithm.

A similar case study analysis was conducted for the retrieval of the characteristic size. The polarimetric $D_m$ algorithm is based on the methodology introduced by Tokay et al. (2020). Much like the situation encountered with rainfall rate ($RR$), the DPR retrieval of $D_m$ often leads to an overestimation of larger drop sizes. This phenomenon can be attributed to the assumption made about the shape of the drop size distribution (DSD). The algorithm operates under the assumption of a constant value for the shape parameter $\mu$, which is fixed at 3. However, recent studies by Williams et al. (2014) and Protat et al. (2019) have demonstrated that, statistically, the shape parameter decreases as the characteristic size increases. Consequently, relying on a constant value for $\mu$ may not be suitable and can result in inaccurate representations of the DSD, ultimately leading to biases in reflectivity simulations. It is important to note that, once again, the OE algorithm was not applied to columns with the largest sizes. However, it is evident that the range of retrieved sizes is smaller compared to the DPR product, and it aligns better with the ground-based algorithm.

To conduct a more comprehensive assessment of the retrievals, we analyzed data spanning an entire year, specifically from 2016. In this analysis, we exclusively focused on DPR columns within the MS swath. Furthermore, all the profiles selected for this study exhibited the presence of the bright band. These criteria were established to ensure that both algorithms were applied to the same set of profiles, enabling a fair and direct comparison between the OE and DPR methodologies. We utilized data from the GPM ground validation program in version 2.4 for this exercise. This dataset is independent of the one that was used to derive a priori statistics on $D_m$, $PR$ and $\sigma_m$, as they were derived using the data from 2015.

The performance of both the DPR and OE rainfall rate retrieval methods exhibits clear similarities. In direct comparison with the polarimetric rainfall products, both approaches tend to exhibit certain systematic biases. They both have a tendency to overestimate small rainfall rates while simultaneously underestimating high rainfall rates. However, within the range of precipitation rates specified by mission requirements, i.e., between 1 and 10 mm h$^{-1}$, some differences between the algorithms emerge.

Within this range of precipitation rates, our retrieval method demonstrates somewhat improved statistical performance. This superiority is characterized by a smaller mean error, which arises from the counterbalancing effect of underestimation for larger precipitation rates and overestimation for smaller ones. In essence, our product achieves a more balanced representation within this precipitation rate range, leading to an improvement in overall accuracy of the rainfall accumulation estimates.

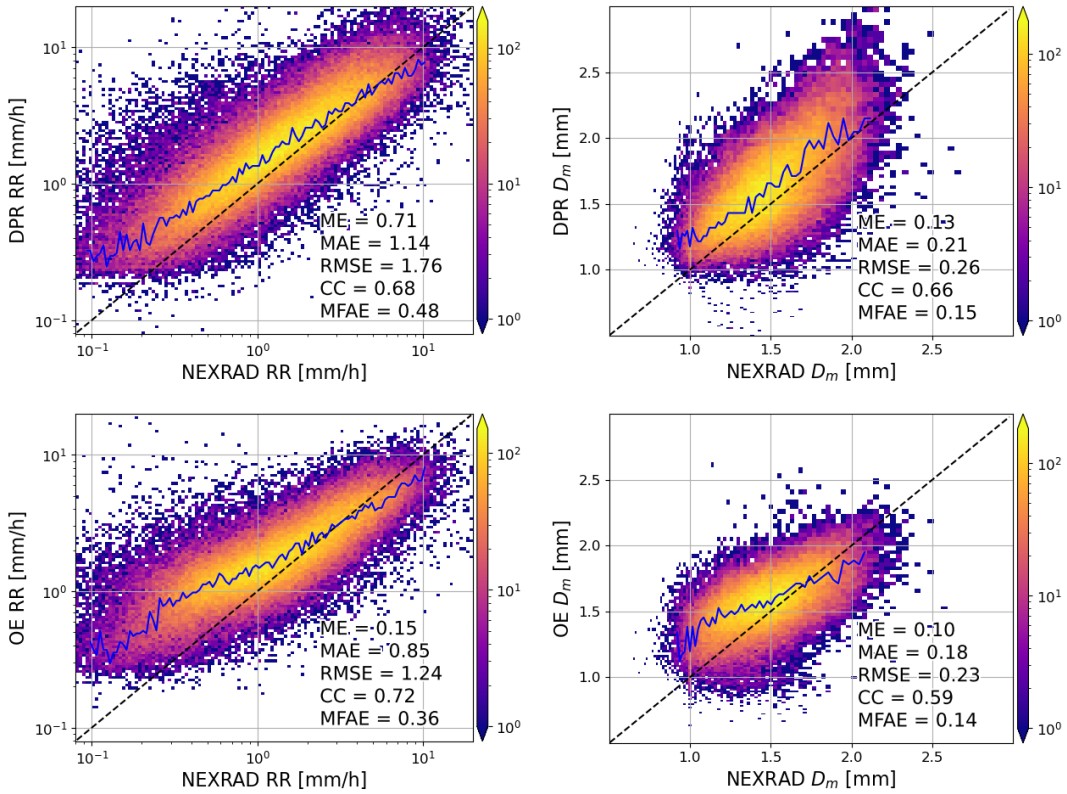

**Figure 11.** One year of statistics for the DPR and OE algorithms compared to the collocated polarimetric radar product. The left column represents the precipitation rate product, while the right column displays the characteristic size estimate. The top row represents the DPR V06 product while the bottom one corresponds to the OE algorithm. Statistical quantities are computed for radar volumes where the precipitation rate falls within the range of 1 to 10 $\mathrm{mm\,h^{-1}}$.

In terms of statistical metrics, our product exhibits slightly improved performance with a smaller mean absolute error and lower root mean square error, measuring at 0.3 and 0.52 $\mathrm{mm\,h^{-1}}$, respectively. While these improvements are notable, it is important to emphasize that they do not represent an impactful deviation from previous methods. Moreover, our product overestimates low rain rates more than the official algorithm. One of the key indicators demonstrating our product's improved performance is the mean fractional absolute error, which has seen a decrease from 0.48 to 0.36. This reduction corresponds to a

12-percentage-point improvement in the fractional error. These results highlight the effectiveness of our approach in providing rainfall rate estimates that match or even exceed the performance of the official GPM product.

The retrieval of the characteristic size exhibits patterns akin to those observed in precipitation rate retrievals. In both the DPR and OE products, there is a tendency to overestimate smaller sizes while underestimating larger ones. This phenomenon is likely a consequence of maintaining a constant shape parameter assumption within the drop size distribution parametrization. Given

that the shape parameter typically decreases as size increases (Williams et al., 2014), the persistence of a constant value may

introduce biases into the retrieval process. In the case of the OE algorithm, this behavior can be attributed to the inherent nature of Bayesian algorithms. Firstly, they narrow the range of retrieved parameters by penalizing large deviations from the statistical mean. Secondly, this penalty is more pronounced for smaller sizes, which correspond to lower reflectivity values characterized by higher uncertainties. Consequently, these smaller reflectivity values contribute less to the final estimate, reinforcing the observed overestimation of smaller characteristic sizes.

When we narrow our statistical analysis of the $D_m$ retrieval to radar volumes falling within the predetermined mission requirements for precipitation rates, we notice a marginal enhancement in error metrics when compared to the DPR product. Specifically, the mean error, mean absolute error, and root mean square error all exhibit a reduction of $0.03\,\mathrm{mm}$, suggesting a slight improvement in accuracy. Additionally, the mean fractional absolute error sees a modest decrease of 1 percentage point, although it is important to note that the correlation coefficient experiences a marginal decline of 0.07. These observations underscore the fact that the proposed methodology effectively matches the performance of the DPR retrieval, demonstrating its reliability within the specified mission requirements.

## 7  Conclusions

This paper introduces a novel algorithm designed for the retrieval of microphysics information within cold rain precipitation regions. Several findings and insights are unveiled throughout the course of this study.

Firstly, the algorithm effectively transfers microphysics data from the rain phase to the snow phase situated above the melting layer. This approach ensures a physically consistent retrieval of precipitation properties, which is a critical advancement in comparison with the Dual-frequency Precipitation Radar (DPR) operational product. Furthermore, the incorporation of variable ice density within the algorithm emerges as a crucial factor in improving the agreement between radar simulations and observations from the DPR, especially in regions adjacent to the melting zone. This aspect of the algorithm increases the likelihood that the derived physical properties align more closely with real-world conditions, further ensuring its accuracy.

The study reveals a tendency of the OE method to overestimate rain size for small rain drops, shedding light on a limitation in the retrieval of rain characteristics that warrants further investigation and refinement. It is expected that this issue is related to the way the Bayes statistics work. Small rain drops occur for low reflectivity measurements that are characterised by higher uncertainty. This increases relative importance of the a-priori assumptions and the algorithm drifts toward statistical mean of the precipitation characteristics rather than fitting the measurements. Moreover, due to sensitivity of the Ka-band system, in the low reflectivity regions, our retrieval is often limited to single frequency data, which may result in poorer quality of the product. The algorithm is still in a development stage, and this paper serves as a proof of concept only. The influence of a universal a-priori assumption on the accuracy of rain rate estimations, particularly in low reflectivity conditions is still investigated and intensity dependent approaches as in Tridon et al. (2019); Battaglia et al. (2015) are under consideration.

While recognizing the significance of matching the performance of the DPR product in rain regions, the primary emphasis of this study is directed towards the retrieval of ice-phase characteristics above the melting zone, which remains a limiting uncertainty in simulations of weather and climate (Sullivan and Voigt, 2021). The development of this novel algorithm not

only achieves similar performances with existing approaches in retrieving rain properties but also offers a more physically consistent retrieval of the continually evolving characteristics of precipitation within the melting zone. This advancement opens new possibilities for quantifying ice density, particularly in the proximity of the freezing level, which holds important implications for our understanding and prediction of mixed-phase precipitation phenomena.

In this study, we deliberately omitted the consideration of multiple scattering in the radar observations at the Ka band for two primary reasons. Firstly, this simplification was implemented to enhance the algorithm's computational speed, making it more efficient for practical use. Secondly, it is worth noting that multiple scattering events occur in highly unusual circumstances, particularly in convective precipitation, as documented in previous research (Battaglia et al., 2014). These events occur when the attenuation coefficient exceeds the reciprocal of the radar beam width, and the scattering albedo is notably high (Battaglia et al., 2010). For the DPR system, such conditions typically arise within the ice phase when precipitation rates surpass $12 \mathrm{~mm~h^{-1}}$ for graupel and $40 \mathrm{~mm~h^{-1}}$ for aggregates of dendrites. As these conditions are rare in cold rain, we deemed them unlikely to impact the majority of precipitation scenarios. However, it is essential to acknowledge that should the need arise, these effects can be accounted for by incorporating the methodology developed by Hogan and Battaglia (2008).

In order to enhance radar performance analysis, the authors employ ground-based data to develop a Non-Uniform Beam Filling (NUBF) radar simulator. Present efforts primarily concentrate on simulating radar reflectivity and attenuation at both the Ku and Ka bands, matching the resolution of the ground-based radar stations. These simulations, in conjunction with microphysical rain properties, are averaged over the DPR footprint size, resulting in a comprehensive radar simulator tailored to target heterogeneous scenes. It's important to acknowledge that this approach does not consider the potential shadowing effect in the attenuation field, as described in Short and Iguchi (2011), and is equivalent to a complete decorrelation of precipitation fields at various levels. Nevertheless, it represents an initial step in addressing the NUBF problem.

In the future development of our algorithm, we are contemplating the strategic utilization of the synergy between space-borne and ground-based measurements to enhance both spatial and vertical resolution. Rather than pursuing an approach focused on upscaling the resolution of radar volumes, our vision centers on refining the retrieval process by harnessing the distinct advantages offered by these two complementary systems. While one system excels in delivering precise vertical resolution, the other boasts exceptional horizontal resolution. Our objective is to leverage this harmonious pairing to elevate our retrieval capabilities, ultimately enabling us to derive precipitation quantities at a finer scale. This forward-looking strategy holds the potential to improve the accuracy and precision of our precipitation retrievals, pushing the boundaries of what is achievable in the fields of remote sensing and meteorology.

The realization of this enhancement necessitates a precise collocation of measurements in both space and time. While spatial collocation poses minimal challenges, temporal alignment remains a considerable concern. Ground-based radar volume scans typically span approximately five minutes, introducing temporal discrepancies. Until the advent of rapid scans from phased array radars becomes available, our precipitation model employed in the retrieval must accommodate factors like advection. This entails assimilating data from multiple radar volumes into the algorithm or implementing corrections for system movements, such as utilizing Optical Flow techniques (Pulkkinen et al., 2019). By integrating these advancements into the Optimal Estimation process, we can establish pseudo triple-frequency retrievals, offering exceptional potential as validation

datasets. Moreover, these retrievals can serve as valuable training data for upcoming radar missions, such as Tomorrow.io, that

rely on data-driven machine learning approaches. This collaborative approach holds the promise of advancing the accuracy and effectiveness of our precipitation retrieval methods, marking a pivotal step in meteorological research and remote sensing technology.

*Code availability.* The Optimal Estimation algorithm along with the scattering tables used in this study is publicly available as a GitHub repository (https://github.com/mrozkamil/gpym).

*Data availability.* The database of single scattering properties of rimed aggregates can be found at https://zenodo.org/record/7510186. GPM data is freely available subject to registration from NASA servers (https://arthurhouhttps.pps.eosdis.nasa.gov/). The GPM ground validation data was obtained from https://pmm-gv.gsfc.nasa.gov/pub/gpm-validation/data/gpmgv/netcdf/geo_match/GPM/2ADPR.

*Author contributions.* KM developed the retrieval algorithm, led writing the manuscript, and prepared all figures. AB assisted in developing the methodology and provided editorial comments. AMF contributed to the scientific discussion and provided an initial draft of the paper.

*Competing interests.* Authors declare no competing interests.

*Acknowledgements.* Work done by Kamil Mroz and Alessandro Battaglia was performed under a contract with the National Centre for Earth Observation. This research used the ALICE High Performance Computing facility at the University of Leicester.

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
