# Peer review of "Enhancing Consistency of Microphysical Properties of Precipitation across the Melting Layer in the Dual-Frequency Precipitation Radar Data"

_EGUsphere, 2023_

## Referee Comment (RC1)

**Enhancing Consistency of Microphysical Properties of Precipitation across the Melting Layer in the Dual-Frequency Precipitation Radar Data**

Mroz et al. 2023 submitted to AMT

**Review by:**

Anonymous Reviewer

**Introduction and Recommendation:**

Spaceborne radars provide unprecedented observations of the 3-dimensional structure of clouds and precipitation. The first meteorological spaceborne radar, named the Tropical Rainfall Measuring Mission (TRMM), was launched back in 1997, and enabled the first estimates of near surface precipitation rates across the global tropics. Through the years of TRMM, an emphasis was put on making the near surface rain rates more accurate. This emphasis carried through to the follow up mission, named the Global Precipitation Measurement Mission (GPM), which launched in 2014. The GPM mission extended spaceborne radar observations to higher latitudes with its more included orbit (65S – 65N).

The authors of the submitted manuscript have sought out to create a new radar retrieval specifically designed for the GPM Dual-Frequency Precipitation Radar and in stratiform precipitation. They show that one of the stock GPM algorithms (2A.DPR) do not have a physically consistent retrieval of precipitation rate through the melting layer (0 degC) which has been discussed in previous literature as a quasi-conserved parameter. Their main conclusions are:

1) Show the deficiency in the 2A.DPR algorithm for consistent retrievals across the melting layer.
2) Provide a new, optimal estimation, retrieval of DSD parameters.
3) Compare the new retrieval to ground based retrievals of the same DSD parameters.

The overall writing is good, and the paper fits the scope of AMT. I do have the following **major comments** that need to be addressed before publication.

**Major comments:**

*Other GPM retrievals:*
The author's center their discussion on the 2A.DPR algorithm, which is a primary product of GPM through the JAXA team. This algorithm uses a R-Dm-Ze relationship to retrieve the DSD (Seto) and the R-Dm relationship for snow is likely inappropriate (Chase et al. 2020). This is acceptable, but given the authors are providing a new retrieval, there should be some discussion around other published retrievals. The main other retrievals that come to mind are:

1) NASA GPM retrieval named the 'Combined' algorithm, which is first discussed in the literature by Grecu et al. (2016). The NASA CMB algorithm is an optimal estimation retrieval and might not have the same deficiencies as the 2A.DPR algorithm. The data are

freely available from the same website as the 2A.DPR files and potentially could be added into the analysis

2) The Chase et al. (2021) neural network retrieval. The focus of the Chase et al. (2021) retrieval was to correct for potential deficiencies in the 2A.DPR algorithm for snowfall (noted in Chase et al. 2020) and showed how the new retrieval compared to CloudSat in Chase et al. (2022). Furthermore, Chase et al. (2022; c.f., Figure 3), showed and discussed how the new neural network retrieval of snowfall rate, matches well with the 2A.DPR rain rate just below the melting level.

I know that adding in new datasets is cumbersome and is not needed for this paper to be published, but at a minimum there needs to be discussion of these two other GPM algorithms and the caveat that the issue noted in the 2A.DPR algorithm might not extend to the others.

**Minor Comments:**

**Focus on Snow to rain transition**
The whole paper has an emphasis on the snow to rain transition, yet the only evaluations done of the new algorithm is on surface rain (section 6). I get why this is done, to show that the new algorithm still gets sufficient rain accuracy, and the rain algorithms are generally better than snow algorithms (at least from NEXRAD). A suggestion here is reproduce Figure 3 with the new algorithm. This would really tie the point home that the consistency across the melting layer has been improved. Ideally this would be shown prior to the bulk evaluation (Figure 7). Can the authors also be explicit that the evaluations in section 6 are near the surface?

**More details on the 2A.DPR algorithm:**
It would be helpful to readers to have a bit more intuition of the 2A.DPR algorithm. For example, noting that it is an R-Dm retrieval, is helpful to provide context to the reader that the algorithm was developed for rain, not snow, and might be the main reason for the discrepancy the authors are highlighting in the manuscript. It would be good to cite the paper that describes the algorithm as well (Seto et al. 2021).

**Length of record:**
Why just 5 years of data? Why not use all of it (2014 – 2023).

**Line by Line comments:**

Note, word suggestions are *suggestions*. Please feel free to disagree.

Line 22: I have seen decent signal of the KuPR down to 12 dBZ. I know that this is not citable in a publication, but just a note.

Line 35: There is a better citation for the Conv/Stratiform retrieval: Awaka et al. (2021)

Line 52: Maybe the word 'stratiform rain volume' is better than 'stratiform rain deck'

Figure 1 caption: Which ray is this? Is it near nadir I assume?

Line 84: I know that aggregates have large non-Rayleigh effects, but is this common knowledge? Should you cite an example here?

Lines 93 – 96: it might be good to mention here that prior to May 2018, there was no matched Ka-band in the outer swath anyway. Making the identification of the bright band harder and no Ka-band for the dual-frequency retrieval anyway.

Line 99: This would be a good spot for the Le and Chandrasekar (2013) reference.

Section 3: This is where some added discussion on the R-Dm retrieval in the 2A.DPR product would be helpful (Seto et al. 2021). Furthermore, it might be good to mention Chase et al. (2020) which evaluated the R-Dm relationships in rain and snow.

Line 174: Can you add 2A.DPR in parenthesis after the V06? This would help folks who know more about the DPR algorithms what files you are using.

Lines 178 – 181: This was noted previously by Chase et al. (2021; c.f., Figure 15).

Lines 212 – 213: The Skofronick-Jackson et al. (2019) and the Casella et al. (2017) papers also documented the snowfall rate deficiency of the 2A.DPR algorithm.

Line 375 – 376: There is a good reference by Heymsfield et al. (2018) that talks about the relative humidity across the melting layer.

Line 463: Refrain from using the word 'significant' unless a statistical test is used. If there was a statistical test used for hypothesis testing, be explicit which ones and what level of significance was used.

Line 464: Suggest switching the order of rain and snow to follow a top-down (i.e., snow falling and melting to rain).

Code Availability: It would be nice to have a simple script to show how to run the OE retrieval developed in this paper. That way readers could run the suggested physically consistent retrieval for their respective scientific endeavors.

**References:**

Jun Awaka , Minda LE , Stacy BRODZIK , Takushi Kubota , Takeshi Masaki , V. CHANDRASEKAR , Toshio Iguchi , Development of a precipitation type classification algorithm for the full scan mode of the dual-frequency precipitation radar of the Global Precipitation Observation Plan, Meteorological Journal. Vol. 2 , 2021 , Volume 99 , Issue 5 , p. 1253-1270 , https:// doi.org/10.2151/jmsj.2021-061

Daniele Casella, Giulia Panegrossi, Paolo Sanò, Anna Cinzia Marra, Stefano Dietrich, Benjamin T. Johnson, Mark S. Kulie, Evaluation of the GPM-DPR snowfall detection capability: Comparison with CloudSat-CPR, Atmospheric Research, Volume 197, 2017, Pages 64-75, https://doi.org/10.1016/j.atmosres.2017.06.018

Chase, R.J.; Nesbitt, S.W.; McFarquhar, G.M. Evaluation of the Microphysical Assumptions within GPM-DPR Using Ground-Based Observations of Rain and Snow. *Atmosphere* **2020**, *11*, 619. https://doi.org/10.3390/atmos11060619

Chase, R. J., S. W. Nesbitt, and G. M. McFarquhar, 2021: A Dual-Frequency Radar Retrieval of Two Parameters of the Snowfall Particle Size Distribution Using a Neural Network. *J. Appl. Meteor. Climatol.*, **60**, 341–359, https://doi.org/10.1175/JAMC-D-20-0177.1.

Chase, R. J., S. W. Nesbitt, G. M. McFarquhar, N. B. Wood, and G. M. Heymsfield, 2022: Direct Comparisons between GPM-DPR and CloudSat Snowfall Retrievals. *J. Appl. Meteor. Climatol.*, **61**, 1257–1271, https://doi.org/10.1175/JAMC-D-21-0081.1.

Grecu, M., W. S. Olson, S. J. Munchak, S. Ringerud, L. Liao, Z. Haddad, B. L. Kelley, and S. F. McLaughlin, 2016: The GPM Combined Algorithm. *J. Atmos. Oceanic Technol.*, **33**, 2225–2245, https://doi.org/10.1175/JTECH-D-16-0019.1.

Heymsfield, A., A. Bansemer, N. B. Wood, G. Liu, S. Tanelli, O. O. Sy, M. Poellot, and C. Liu, 2018: Toward Improving Ice Water Content and Snow-Rate Retrievals from Radars. Part II: Results from Three Wavelength Radar–Collocated In Situ Measurements and CloudSat–GPM–TRMM Radar Data. *J. Appl. Meteor. Climatol.*, **57**, 365–389, https://doi.org/10.1175/JAMC-D-17-0164.1.

Kota Seto , Toshio Iguchi , Robert MENEGHINI , Jun Awaka , Takashi Kubota , Takeshi Masaki , Nobuhiro Takahashi , Precipitation intensity estimation algorithm for the Global Precipitation Observation Plan Dual-frequency Precipitation Radar (GPM/DPR), Meteorological Journal. Vol. 2 , 2021 , Volume 99 , Issue 2 , p. 205-237 , https://doi. org/10.2151/jmsj.2021-011

Skofronick-Jackson, G., M. Kulie, L. Milani, S. J. Munchak, N. B. Wood, and V. Levizzani, 2019: Satellite Estimation of Falling Snow: A Global Precipitation Measurement (GPM) Core Observatory Perspective. *J. Appl. Meteor. Climatol.*, **58**, 1429–1448, https://doi.org/10.1175/JAMC-D-18-0124.1.

---

## Author Response (AR1)

On behalf of all co-authors, I want to extend our gratitude for reviewing our paper and offering valuable comments. Your insights significantly enhanced the depth of the presented analysis and pointed out deficiencies in the original manuscript. In the subsequent response, I will address each comment point by point, using red font to facilitate easy identification.

Reviewer 1:

**Enhancing Consistency of Microphysical Properties of Precipitation across the Melting Layer in the Dual-Frequency Precipitation Radar Data**

Mroz et al. 2023 submitted to AMT

**Review by:**

Anonymous Reviewer

**Introduction and Recommendation:**

Spaceborne radars provide unprecedented observations of the 3-dimensional structure of clouds and precipitation. The first meteorological spaceborne radar, named the Tropical Rainfall Measuring Mission (TRMM), was launched back in 1997, and enabled the first estimates of near surface precipitation rates across the global tropics. Through the years of TRMM, an emphasis was put on making the near surface rain rates more accurate. This emphasis carried through to the follow up mission, named the Global Precipitation Measurement Mission (GPM), which launched in 2014. The GPM mission extended spaceborne radar observations to higher latitudes with its more included orbit (65S – 65N).

The authors of the submitted manuscript have sought out to create a new radar retrieval specifically designed for the GPM Dual-Frequency Precipitation Radar and in stratiform precipitation. They show that one of the stock GPM algorithms (2A.DPR) do not have a physically consistent retrieval of precipitation rate through the melting layer (0 degC) which has been discussed in previous literature as a quasi-conserved parameter. Their main conclusions are:

1. 1) Show the deficiency in the 2A.DPR algorithm for consistent retrievals across the melting layer.
2. 2) Provide a new, optimal estimation, retrieval of DSD parameters.
3. 3) Compare the new retrieval to ground based retrievals of the same DSD parameters.

The overall writing is good, and the paper fits the scope of AMT. I do have the following **major comments** that need to be addressed before publication.

**Major comments:**

*Other GPM retrievals:*

The author's center their discussion on the 2A.DPR algorithm, which is a primary product of GPM through the JAXA team. This algorithm uses a R-Dm-Ze relationship to retrieve the DSD (Seto) and the R-Dm relationship for snow is likely inappropriate (Chase et al. 2020). This is acceptable, but given the authors are providing a new retrieval, there should be some discussion around other published retrievals. The main other retrievals that come to mind are:

1) NASA GPM retrieval named the 'Combined' algorithm, which is first discussed in the literature by Grecu et al. (2016). The NASA CMB algorithm is an optimal estimation retrieval and might not have the same deficiencies as the 2A.DPR algorithm. The data are freely available from the same website as the 2A.DPR files and potentially could be added into the analysis

2) The Chase et al. (2021) neural network retrieval. The focus of the Chase et al. (2021) retrieval was to correct for potential deficiencies in the 2A.DPR algorithm for snowfall (noted in Chase et al. 2020) and showed how the new retrieval compared to CloudSat in Chase et al. (2022). Furthermore, Chase et al. (2022; c.f., Figure 3), showed and discussed how the new neural network retrieval of snowfall rate, matches well with the 2A.DPR rain rate just below the melting level.

I know that adding in new datasets is cumbersome and is not needed for this paper to be published, but at a minimum there needs to be discussion of these two other GPM algorithms and the caveat that the issue noted in the 2A.DPR algorithm might not extend to the others.

Indeed, the primary focus of the paper is on the DPR product, given that our retrieval framework relies on radar-only measurements. Furthermore, in a previous study where we compared various GPM snowfall products over the continental US, we observed minimal differences between radar-only products and the radar-radiometer combined algorithm (CORRA). Since we had already acquired the CORRA product for other applications, we conducted an analysis on the mass flux and $D_m$ changes within the melting zone for this algorithm as well.

It came as no surprise to us that the radar-radiometer product is also impacted by the same issue of deficiency in the precipitation rate above the melting zone. In fact, the relationship between rainfall and snowfall rate in this product closely resembles that of the DPR product, as illustrated in the figures below:

[Figure]

The following discussion was added to the paper:

"We conducted a similar analysis with another official GPM product, specifically the 2B.GPM.DPRGMI.CORRA. This algorithm integrates DPR data with passive measurements from the GPM Microwave Imager (GMI). We evaluated the same version of this product over a consistent time span of 5 years. The findings indicated that this product is also subject to the same issue—specifically, the precipitation rate above the melting layer is only one-third of the rainfall rate below the melting zone. The regression line PR (ice) = 0.29 PR(rain) + 0.89 best describes the mass flux change between the phases. Similar to the radar-only product, the melted equivalent size of precipitation is well preserved through the melting zone ($D_m$ (ice) = 0.83 $D_m$ (rain) + 0.14).

The striking similarities between the two official DPR products were already observed by Mroz at al. (2021), who noted that both products are affected by a similar underestimation of snowfall rates over the continental US. For brevity and given the similarities to the corresponding figures for the DPR product, the histograms illustrating these results are not included in this paper."

Regarding the alternative product, we obtained the neural network retrieval developed by Chase et al. (2021) from the repository and applied it to measurements above the melting zone. This analysis was performed with profiles where our OE algorithm was executed, i.e. for the DPR profiles within 100 km from the NEXRAD radars. Anticipating an alignment in precipitation rates below and above the melting zone in the NN retrieval, we conducted this exercise to assess how well the AI algorithm preserves the mass-weighted melted equivalent size of precipitation during the melting process.

To our surprise, the algorithm by Chase et al. (2021) also exhibited a noticeable deficiency in precipitation rates above the melting zone. The histograms depicting this behaviour are presented below:

[Figure]

To understand why the NN retrieval is influenced by this issue, we generated a joint probability density function (PDF) of precipitation rates as retrieved in ice by the DPR algorithm and the NN product (see below). The plot indicates that small precipitation rates are much larger in the NN algorithm. However, for large mass flux the two products appear

to converge to the same value. Hence, the transition from ice to rain is characterized by a substantial increase in the NN algorithm mainly for large precipitation rates. We believe that an in-depth intercomparison of these products could potentially be explored in another publication, given that it would necessitate more comprehensive efforts.

[Figure]

The following discussion was added in the text to stress the need for more intercomparison studies:

"Notably, our product improves the coherence between precipitation rates above and below the melting zone. Similar to the DPR product, there is an increase in water mass flux during the transition from ice to rain, but in our case, this increase is more subdued. Upon examining the regression lines, the expected snowfall rate above the melting zone, with 10 mm/h of rainfall underneath, equals 6 mm/h for our product and 3 mm/h for the DPR product.

Overall, the mean fractional bias (MFB), defined as:

$$MFB = \exp\left[\frac{1}{N}\sum_{i=1}^{N}\ln\left(PR_{rain}^{i}\right) - \ln\left(PR_{ice}^{i}\right)\right] - 1,$$

is equal to 4% and 106% for the OE and the DPR algorithms, respectively. Using DPR precipitation rates in rain as the reference for our product, the snowfall rate is on average 30% smaller than the rainfall rate underneath.

In the validation study conducted by Chase et al. (2022), it was demonstrated that the Neural Network snowfall algorithm (Chase et al.,2021), designed specifically for the DPR, exhibits significantly improved agreement between snow and ice phase precipitation rates compared to the DPR product. Their algorithm almost perfectly aligns with the mass flux between the phases, showing only a 2% difference. This underscores the remarkable success achievable with artificial intelligence algorithms when trained using the right database. However, it is crucial to note that its accuracy is contingent on the precision of the attenuation correction, particularly at the Ka-band, which is more susceptible, especially in heavy precipitation conditions.

While our algorithm doesn't achieve a perfect precipitation rate match between different water phases, it brings about noticeable improvement compared to the DPR product. The algorithm's development is still in its early stages, and ongoing adjustments in the code are expected to reduce this discrepancy. Preliminary tests have been conducted, revealing that the continuity of the mass flux through the melting zone can be enhanced at the expense of the continuity of the melted equivalent size. However, it remains debatable which of these two properties exhibits less variability within the melting zone, and this should be a topic for future studies.

Furthermore, it's important to acknowledge that some changes in the precipitation rate may still occur within the melting zone, resulting in an anticipated growth in precipitation rate. The extent of this change depends on environmental conditions such as relative humidity and temperature profiles (Heymsfield et al., 2018)."

**Minor Comments:**

*Focus on Snow to rain transition*

The whole paper has an emphasis on the snow to rain transition, yet the only evaluations done of the new algorithm is on surface rain (section 6). I get why this is done, to show that the new algorithm still gets sufficient rain accuracy, and the rain algorithms are generally better than snow algorithms (at least from NEXRAD). A suggestion here is reproduce Figure 3 with the new algorithm. This would really tie the point home that the consistency across the melting layer has been improved. Ideally this would be shown prior to the bulk evaluation (Figure 7). Can the authors also be explicit that the evaluations in section 6 are near the surface?

Thank you for your insightful comment. We have incorporated the suggested Figure 3, which explicitly illustrates the improved consistency across the melting layer using the new algorithm. This figure is now included before the bulk evaluation (Figure 7).

Regarding the validation of the product in rain, we acknowledge the challenges associated with validating the algorithm in ice due to the limited number of validation underflights within stratiform precipitation. Even when these flights are conducted, they are often characterized by discrepancies in sampling time and significant differences in sampling volume due to the high ground track speed of the satellite. For instance, an in situ aircraft flying at 600 km/h only crosses over two DPR pixels in one minute. This means that within a 10-minute window, we only collect approximately 20 validation points, which may not adequately represent the entire radar volume due to their proximity to a one-dimensional cut through a 5x5x0.25 km³ volume.

Our validation in rain, was performed with all radar volumes below the melting zone and we added this statement in the paper to make it clear.

The following discussion was also added:

"An example of the OE retrieval is presented in Fig. 7. The rain component aligns with the overall structure observed in the DPR product, featuring distinct elements of a squall line system: a convective core, a transition zone, and stratiform precipitation, viewed from east to west. However, the range of retrieved parameters is smaller compared to the DPR algorithm,

even within stratiform rain profiles. This discrepancy is mainly caused by differences in the differential attenuation fitting between the two algorithms. The DPR product tends to overestimate this parameter as it tries to compensate for the non-uniform beam filling effect, while our product does not incorporate this correction. It must be acknowledged that the reliability of the OE product is questionable in convective precipitation, given its design for stratiform precipitation. Consequently, the presence of artifacts at approximately 5 km altitude (a peak in $D_m$ and PR) within the convective core should not come as a surprise.

In the ice phase, the OE algorithm tends to yield a larger precipitation rate than the DPR product. Both the PR and $D_m$ reach their peaks above the melting zone and remain relatively constant in rain; that is, the majority of the growth occurs within the ice phase. Moreover, compared to the DPR product, the precipitation structure in the ice phase is not affected by strong discontinuities during the transition from one profile to the other, although no constraints on horizontal variability were imposed."

Added figure:

[Figure]

*More details on the 2A.DPR algorithm:*

It would be helpful to readers to have a bit more intuition of the 2A.DPR algorithm. For example, noting that it is an R-Dm retrieval, is helpful to provide context to the reader that the algorithm was developed for rain, not snow, and might be the main reason for the

discrepancy the authors are highlighting in the manuscript. It would be good to cite the paper that describes the algorithm as well (Seto et al. 2021).

The following discussion was added: The DPR retrieval algorithm utilizes measured radar reflectivity, total path integrated attenuation estimates corrected for non-precipitating particles, the relationship between precipitation rate and mass-weighted mean diameter (P R − Dm), and phase information based on the melting layer detection. It generates profiles of precipitation rate and drop size distribution parameters (Dm, Nw). Additionally, profiles of effective reflectivity and specific attenuation coefficients are provided. The algorithm employs the P R − Dm relationship with an adjustment parameter, ε, aiming to reconcile discrepancies between the surface reference technique PIA and the one simulated from hydrometeor profiles. Version 06 had a single ε value along the profile, while Version 07 introduces varying ε in the column.

The P R-Dm relation, replaces the traditionally used relation involving specific attenuation (k) and effective radar reflectivity factor (Ze). While using the k-Ze relation with the Hitschfeld-Bordan attenuation correction method (Hitschfeld and Bordan, 1954) enables the derivation of a Ze profile from the Zm profile without the need for scattering tables, this relation is not applicable at the Ka-band due to the weaker correlation between involved parameters. This limitation arises from rain extinction being strongly affected by absorption rather than being dominated by scattering. Consequently, the Hitschfeld-Bordan method leads to inconsistencies in attenuation correction at two frequencies.

The algorithm follows a logical sequence: assuming a gamma DSD with a fixed shape parameter, a relationship between PR and Dm imposes a unique solution for a given effective reflectivity. Consequently, the corresponding values for Nw is found and by using the scattering tables the specific attenuation coefficient k is obtained. The process begins at the top, where the measured reflectivity is assumed to be unaffected by attenuation and is iteratively corrected using the estimated k. This procedure is applied throughout the column, resulting in the attenuation profile. The process is iterated with different values of ε to minimize the difference between the simulated PIA at the SRT-estimate.

For more details about the changes introduced in version 6 of the GPM-DPR algorithm, refer to the Algorithm Theoretical Basis Document (Iguchi et al., 2018) or to the algorithm description provided by SETO et al. (2021). Additionally, the study conducted by Chase et al. (2020) provides a thorough evaluation of the PR-Dm relation in both rain and snow using disdrometer measurements. They conclude that the P R-Dm retrieval may not be optimal in snow due to the variability of snowflake. mass, suggesting the exploration of alternative techniques.

*Length of record:*

Why just 5 years of data? Why not use all of it (2014 – 2023).

Given the extensive size of the complete dataset, we opted to utilize a five-year period of data for our analysis. We are confident that the statistics derived from this five-year period are

sufficiently robust and that incorporating additional data would not significantly alter the findings.

**Line by Line comments:**

Note, word suggestions are *suggestions*. Please feel free to disagree.
Line 22: I have seen decent signal of the KuPR down to 12 dBZ. I know that this is not citable in a publication, but just a note.

We refer to literature values, but we are aware that in many cases, the Ku-band radar reflectivity seems to capture precipitation as weak as 12 dBZ. Nevertheless, the probability density function (PDF) of measured reflectivity peaks at higher values, indicating that some returns below 15 dBZ are missed.

Line 35: There is a better citation for the Conv/Stratiform retrieval: Awaka et al. (2021)

Added

Line 52: Maybe the word 'stratiform rain volume' is better than 'stratiform rain deck'

Changed

Figure 1 caption: Which ray is this? Is it near nadir I assume?

The ray number has been added.

Line 84: I know that aggregates have large non-Rayleigh effects, but is this common knowledge? Should you cite an example here?

We cited Kuo et al. (2016)

Lines 93 – 96: it might be good to mention here that prior to May 2018, there was no matched Ka-band in the outer swath anyway. Making the identification of the bright band harder and no Ka-band for the dual-frequency retrieval anyway.

We added your comment.

Line 99: This would be a good spot for the Le and Chandrasekar (2013) reference.

We cited their paper where you suggested.

Section 3: This is where some added discussion on the R-Dm retrieval in the 2A.DPR product would be helpful (Seto et al. 2021). Furthermore, it might be good to mention Chase et al. (2020) which evaluated the R-Dm relationships in rain and snow.

A more detailed description of the DPR algorithm was added.

Line 174: Can you add 2A.DPR in parenthesis after the V06? This would help folks who know more about the DPR algorithms what files you are using.

Lines 178 – 181: This was noted previously by Chase et al. (2021; c.f., Figure 15).

We cited Chase et al., 2021) here.

Lines 212 – 213: The Skofronick-Jackson et al. (2019) and the Casella et al. (2017) papers also documented the snowfall rate deficiency of the 2A.DPR algorithm.

The following statement was added: Similar problem with the snowfall deficiency in the 2A.DPR product was also reported by Skofronick-Jackson et al. (2019) and Casella et al. (2017).

Line 375 – 376: There is a good reference by Heymsfield et al. (2018) that talks about the relative humidity across the melting layer.

We added this citation.

Line 463: Refrain from using the word 'significant' unless a statistical test is used. If there was a statistical test used for hypothesis testing, be explicit which ones and what level of significance was used.

The word 'significant' was replaced with more appropriate synonyms as no formal statistical tests were conducted. The original usage of 'significant' was more informal and imprecise.

Line 464: Suggest switching the order of rain and snow to follow a top-down (i.e., snow falling and melting to rain).

We kept the original order as the rain retrieval is simpler so the information flows from rain to ice.

Code Availability: It would be nice to have a simple script to show how to run the OE retrieval developed in this paper. That way readers could run the suggested physically consistent retrieval for their respective scientific endeavors.

We've updated the README file in the repository to point to a new Jupyter notebook, demonstrating the step-by-step execution of the retrieval process. This informative notebook is titled "test.ipynb."

**Reviewer 2:**

Major comments:

The OE method is well described, but it is difficult for the audience who are not familiar with GPM DPR algorithms to understand the novelty of this work. Since this work is expected to 'improve' GPM retrievals, the GPM algorithm should be well presented. In particular, the authors should discuss the aspects that this algorithm are different from GPM.

We have provided a more detailed description of the algorithm. In addition, we added a subsection that summarizes main differences and similarities between the algorithms:

In Section 3 we added:

[revised manuscript text omitted]

The manuscript is poor in organization. Section 2 is DPR measurements; Section 3 is DPR retrieval, and 4 for OE. I understand that we usually inference the reason from results. However, it is better to analyze the issues in DPR retrieval and then show the issues in observations in a scientific paper.

We chose to adopt this structure to establish a logical flow, beginning with the presentation of DPR measurements, as they form the foundation of our retrievals. Subsequently, we introduce the official retrieval framework to highlight the issues impacting the precipitation product. The OE algorithm, presented later in the manuscript, is then proposed as a solution to address these identified issues. It's important to note that our intention is not to delve into issues related to the observations. Instead, we adhere to the conventional structure found in AMT articles, where the methodology section logically follows the presentation of data.

Validation should be made in ice. The current 'validation' is sanity check, not validation, since the validation was not in ice. The validation should be made against in-situ measurement of ice. There are several aircraft campaigns designed for GPM validation, and the in-situ observations can be used for quantitative validation.

Concerning the validation of our product in rain, we decided to perform it this way due to the limited number of validation under-flights in the ice phase during stratiform precipitation events. As far as our knowledge extends, only one flight was conducted throughout the entire OLYMPEX campaign, and this event was utilized in the study by Chase et al. (2021). In this study, only a qualitative assessment of their product was performed, refrained from direct comparisons due to disparities in sampling time during in-situ flights and significant differences in sampling volume. The dime difference is caused by the high ground track speed of the satellite. For instance, an in-situ aircraft traveling at 600 km/h intersects only two DPR pixels in one minute. Within a 10-minute window, approximately 20 validation points are collected. This prompts a crucial question regarding the representativeness of the sample and the robustness of potential statistical comparisons. Moreover, in-situ sampling is potentially insufficient to adequately represent the entire radar volume, given their proximity to a one-dimensional cut through a 5x5x0.25 km³ volume. The impact of this sampling volume difference could potentially be mitigated with the collection of large statistics, as discrepancies in the sampling volumes would result in random noise only. However, that would require a lot of flights. Chase et al. (2021) utilized airborne radar data at finer horizontal and vertical resolution for more robust statistics. While we acknowledge their efforts, it's crucial to note that airborne data differ significantly from spaceborne measurements. Airborne data exhibit superior sensitivity, resolution, and reduced signal fluctuations. Additionally, they are less affected by non-uniform beam filling effects compared to satellite measurements.

As you rightly noted, the validation section in our study primarily served as a sanity check. It showcased that a more physically consistent retrieval could be attained without compromising the integrity of the rainfall product. Although our proposal for a more extensive in-situ validation study of DPR products was not secured, this article stands as a proof of concept. The retrieval algorithm is publicly available under the MIT license, welcoming exploration by everyone, including the GPM algorithm team.

To address your comment, we have renamed this section to "Performance Assessment in Rain." Additionally, we have included the following discussion at the beginning of the section:

"The validation of the OE algorithm was exclusively conducted within the rainy portion of the radar profiles. This might appear surprising, given the anticipated improvement in algorithm quality

compared to the DPR product above the freezing level. However, this approach is expedient due to the limited availability of DPR under-flights within snow during stratiform precipitation events. To the best of our knowledge, only one flight was conducted throughout the entire OLYMPEX campaign, and this singular event was utilized in the study by Chase et al. (2021).

In their study, only a qualitative assessment of the product was conducted, refraining from direct comparisons due to disparities in sampling time during in-situ flights and significant differences in sampling volume. The discrepancy in sampling time arises from the high ground track speed of the satellite (7 km s$^{-1}$) compared to approximately 600 km h$^{-1}$ of an in-situ aircraft. Consequently, within a 10-minute window, only 20 validation points are collected.

This raises a critical question about the representativeness of the sample and the robustness of potential statistical comparisons. Moreover, in-situ sampling may be inadequate to sufficiently represent the entire radar volume, given its proximity to a one-dimensional cut through a 5×5×0.25 km³ volume. The impact of this difference in sampling volume could potentially be mitigated with the collection of large statistics, as discrepancies in the sampling volumes would result in random noise only.

However, as pointed out earlier, collecting these statistics is impractical due to the limited number of validation points per flight, making such an effort very expensive. Chase et al. (2021) overcame this issue by utilizing airborne radar data at finer horizontal and vertical resolutions for more robust statistics. While we acknowledge their efforts, it's crucial to note that airborne data differ significantly from spaceborne measurements. Airborne data exhibit superior sensitivity, resolution, and reduced signal fluctuations. Additionally, they are less affected by non-uniform beam filling effects compared to satellite measurements.

The validation presented here served as a sanity check, aiming to assess whether a physically consistent retrieval could be achieved without compromising the integrity of the DPR rainfall product."

**Technical comments (in order of their appearance in manuscript):**

Eq. 11-15: How did you get these parameterizations?

We derived them from derived from the polarimetric radar retrieval. We have made it clear in the text now.

Eq. 20: Where is this equation from?

This is a change of coordinates formula demonstrating how to convert a vector from the space of principal components into a Cartesian coordinate system. In addition to the standard conversion, there is also a normalization step. We included this formula for a more technical audience actively involved in the development of these algorithms.

Line 366: in precipitation properties and and evaluating the accuracy. Extra 'and'

Repetition was removed.

Figure 8 is presented in a single-row, side-by-side arrangement, yet the caption indicating 'Top Panel' and 'Bottom Panel. This discrepancy may be attributed to formatting issues, and it warrants verification.

It was due to formatting, thank you for pointing it out.

Line 413: A similar, case study analysis - A similar case study analysis

The comma was removed.

Line 475: rater than – rather than

It's corrected now.

---

## Author Response (AR2)

Dear Reviewer,
I appreciate your invaluable comments, and I would like to express my gratitude for the insights you provided. Please find our responses to the raised points below, highlighted in red font.

I appreciate the authors for very thorough responses to my concerns. It is easier for me to follow the merits of this work. After reading over this manuscript, I feel there are some vague statemens that may be clarified in the final stage.

1. Equation 18. Matrosov 2008 did not give the atteunation parameterizatoin at Ku-band, how did you get the parameters at Ku band? The used values should be specified in the main text. Also, Li and Moisseev (2019) suggested that Matrosov (2008) overestimates the attenuation, how would this affect your estimates?

Notably, Matrosov's study in 2008 did not encompass simulations at the Ku-band. Consequently, we opted to utilize parameters derived for the X-band. The rationale behind this decision, along with a brief discussion on the work of Li and Moisseev (2019), has been incorporated into the document, presented as follows:
"Here, $\gamma_{ML}$ and $\delta_{ML}$ are wavelength-specific parameters. For the Ka-band, their values are 0.66 and 1.1, respectively. In the case of Ku-band simulations, we adopt the values obtained from X-band simulations, specifically 0.048 for $\gamma_{ML}$ and 1.05 for $\delta_{ML}$. Although we acknowledge that X-band attenuation is likely to be smaller than that of Ku-band, we use it solely as a soft constraint or a priori value. The final attenuation estimate is subsequently refined during the OE iterations. In the study of Li and Moisseev (2019), it was suggested that synthetic simulations by Matrosov tend to overestimate attenuation for snowfall rates exceeding 2.5 mm h$^{-1}$. However, their study was limited to radar measurements exhibiting clear signatures of supercooled clouds above the freezing level. This limitation implies that the study was restricted to rimmed particles only. To accommodate potential variations in the melting layer attenuation estimates, we operate under the assumption that they are subject to a factor of 2 uncertainty (see the next section)."

2. I checked the codes contributed by the authors, and it seems that the raw reflectivity was used. As far as I know, the dual-frequency radars were routinely calibrated by some offset parameters (Awaka et al., 2021). Have you checked that?

The data we employ has already undergone calibration. Based on our observation, the sole correction applied to reflectivity in the study by Awaka et al. (2021) is associated with the attenuation caused by non-precipitating particles and gases. To align with this methodology, we have explicitly included a statement in our article to convey this:
"The vector of measurements consists of the measured values of Z and the differential PIA that are corrected for attenuation by non-precipitating particles and atmospheric gases (Kubota et al., 2020)."